# RECONSTRUCT ANYTHING MODEL:
# A LIGHTWEIGHT GENERAL MODEL FOR COMPUTATIONAL IMAGING

**Matthieu Terris**[1,2]  **Samuel Hurault**[3]  **Maxime Song**[4]  **Julián Tachella**[5,2]

[1]Université Paris-Saclay, Inria, CEA  [2]Blur Labs  [3]ENS Paris, PSL, CNRS
[4]CNRS UAR 851, Université Paris-Saclay  [5]ENSL, CNRS UMR 5672

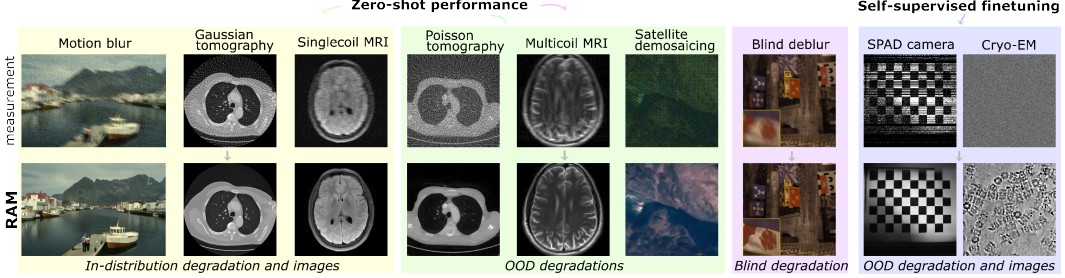

Figure 1: The proposed **R**econstruct **A**nything **M**odel (**RAM**) model can solve a wide variety of inverse problems, from medical imaging to microscopy and low-photon imaging, obtaining state-of-the-art zero-shot performance in imaging problems and datasets in or close to the training distribution. RAM can also be finetuned in a self-supervised way (without any ground-truth references) for images and/or problems strongly differing from training distribution.

## ABSTRACT

Most existing learning-based methods for solving imaging inverse problems can be roughly divided into two classes: iterative algorithms, such as plug-and-play and diffusion methods leveraging pretrained denoisers, and unrolled architectures that are trained end-to-end for specific imaging problems. Iterative methods in the first class are computationally costly and often yield suboptimal reconstruction performance, whereas unrolled architectures are generally problem-specific and require expensive training. In this work, we propose a novel non-iterative, lightweight architecture that incorporates knowledge about the forward operator (acquisition physics and noise parameters) without relying on unrolling. Our model is trained to solve a wide range of inverse problems, such as deblurring, magnetic resonance imaging, computed tomography, inpainting, and super-resolution, and handles arbitrary image sizes and channels, such as grayscale, complex, and color data. The proposed model can be easily adapted to unseen inverse problems or datasets with a few fine-tuning steps (up to a few images) in a self-supervised way, without ground-truth references. Throughout a series of experiments, we demonstrate state-of-the-art performance from medical imaging to low-photon imaging and microscopy. Our code is available at https://github.com/matthieutrs/ram.

## 1 INTRODUCTION

Computational imaging problems are ubiquitous in applications ranging from demosaicing in computational photography to Magnetic Resonance Imaging (MRI). In this work, we focus on linear inverse problems of the form

$$\boldsymbol{y} \sim p(\boldsymbol{y}|\boldsymbol{Ax}), \tag{1}$$

where $\boldsymbol{x} \in \mathbb{R}^n$ is the image to recover, $\boldsymbol{y} \in \mathbb{R}^m$ are measurements, $\boldsymbol{A} \colon \mathbb{R}^n \to \mathbb{R}^m$ is an operator describing the acquisition physics, usually assumed to be linear, and $p$ is the noise distribution,

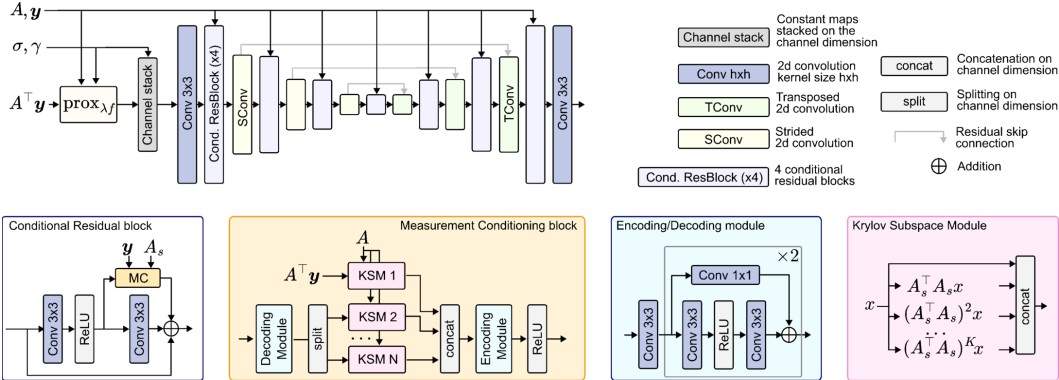

Figure 2: **Proposed architecture for solving non-blind imaging inverse problems.** *Top row:* The architecture builds upon a DRUNet backbone, originally designed with convolutional and residual blocks, but is enhanced to integrate knowledge about the measurement operator $A$ and measurements $y$. *Bottom row:* At each scale, feature maps are decoded into the image domain, processed through a Krylov subspace module (KSM), and then re-encoded. The encoding/decoding module consists of a simple residual convolutional block. The KSM blocks concatenate power iterations of the scaled measurement operator $A_s^\top A_s$, enabling efficient and adaptable processing for a wide range of inverse problems.

which can typically be Gaussian and/or Poisson noise. We mostly focus on non-blind problems with known $A$.

The most common approach to solving the inverse problem (1) is to employ an iterative algorithm to minimize an objective function of the form

$$\arg\min_{x} \frac{1}{2}\|Ax - y\|_2^2 + g(x), \tag{2}$$

where $g$ encodes prior knowledge about the data. Classical approaches rely on handcrafted priors such as total variation (Rudin et al., 1992) or wavelet sparsity (Mallat, 2009). However, recent advances have highlighted the effectiveness of learned priors, particularly deep denoising neural networks, which have been successfully integrated into iterative reconstruction methods. These include Plug-and-Play (PnP) algorithms (Venkatakrishnan et al., 2013; Reehorst & Schniter, 2018; Pesquet et al., 2021; Hertrich et al., 2021; Hurault et al., 2021), and the more recent diffusion models-based methods (Chung et al., 2022; Karras et al., 2022; Daras et al., 2024; Zhu et al., 2023). The advantage of these approaches is that a single model trained only for denoising can be used to solve a large set of image restoration tasks. However, these methods come with several major drawbacks: they are often slow, may introduce blur (Blau & Michaeli, 2018; Milanfar & Delbracio, 2024; Terris et al., 2024), and depend heavily on the training data, limiting their use in domains with scarce ground-truth references (Belthangady & Royer, 2019).

Another common strategy for solving (1) is to train an end-to-end deep neural network specifically for the given task. In low-level vision applications such as single-image super-resolution and image denoising, architectures are often derived empirically, with UNet-based models emerging as a standard choice (Zhang et al., 2021; 2023). Notably, the UNet has also become a prevalent backbone for denoising tasks in generative modeling (Song & Ermon, 2019; Rombach et al., 2022; Karras et al., 2024). However, in computational imaging tasks (e.g., MRI, computed tomography, astronomical imaging), unrolled architectures, inspired by optimization algorithms, are typically preferred due to their ability to incorporate the measurement operator, $A$, and observed data (Adler & Öktem, 2018; Hammernik et al., 2018; Ramzi et al., 2022). Yet, the architecture of unrolled models strongly differs from state-of-the-art UNets used for low level tasks. In particular, empirical results (Zhang et al., 2021; 2023; Zbontar et al., 2018; Zamir et al., 2022b) suggest that effective architectures need not strictly adhere to optimization-derived operations. In both cases, however, even minor architectural modifications—such as adjusting the number of input and output channels—often require full model retraining. This reduces adaptability across datasets and inverse problems with similar statistical properties, such as transitioning from grayscale to color images or handling complex multi-channel representations (Liang et al., 2021).

In this work, we propose a novel neural network architecture for solving (1), building on the DRUNet convolutional neural network (Zhang et al., 2021). Our key modification introduces a conditioning mechanism that integrates the acquisition physics $(\boldsymbol{A}, p)$ through multigrid Krylov iterations. We train a single backbone model across multiple imaging tasks simultaneously ranging from image deblurring to MRI, handling datasets with grayscale, color, and complex images, corrupted by both Gaussian and Poisson noise. As a result, our architecture supports grayscale (1 channel), complex-valued (2 channels), and color images (3 channels), with only the output heads differing to accommodate specific channel numbers. Beyond its strong zero-shot performance, we demonstrate that the model can be efficiently fine-tuned in a self-supervised way on real measurement data.

## 2 RELATED WORKS

**End-to-end architectures**  Several architectures have imposed themselves as cornerstone in the low-level vision community. Early breakthroughs include the UNet (Ronneberger et al., 2015; Jin et al., 2017) and deep residual convolutional networks (Zhang et al., 2017a; 2018b; 2017b). More recently, architectures incorporating attention mechanisms have surpassed previous state-of-the-art models (Zhang et al., 2023; Liang et al., 2021; Zamir et al., 2022b). Akin to our work, (Ren et al., 2023) propose to condition a denoising diffusion UNet on a multiscale version of the measurements. Notable recent developments include transformer-based models, particularly in single-image super-resolution (Liang et al., 2021; Zamir et al., 2022b), and, more recently, state-space models (Guo et al., 2024). Despite these advances, the UNet remains a key backbone for many state-of-the-art methods, see e.g. (Zhang et al., 2023; Zamir et al., 2022b), incorporating transformer blocks on UNet-like architectures.

**Unrolled architectures**  To better condition on the acquisition physics, unrolled networks introduce learnable neural modules (e.g., convolutional layers) within the iterations of an optimization algorithm solving (2). Several architectures have become standard in this framework, notably in natural image processing (Zhang et al., 2017b; 2020; Bertocchi et al., 2020; Sanghvi et al., 2022), medical imaging (Adler & Öktem, 2018; Ramzi et al., 2022; Gossard & Weiss, 2024), and astronomical imaging (Aghabiglou et al., 2023). However, these models have large memory footprints and are typically trained on a single or limited set of tasks with limited variations. Moreover, there is a significant discrepancy between the overall architecture of these unrolled models and the highly effective architectures employed by the previously discussed state-of-the-art UNet-like architectures.

**Self-supervised learning**  In many scientific and medical imaging applications, obtaining ground-truth data for supervised training is often expensive or even impossible (Belthangady & Royer, 2019). Self-supervised methods enable training directly from measurements alone (Batson & Royer, 2019; Chen et al., 2021; Tachella et al., 2023b). In this work, we show that self-supervision is highly effective for finetuning a foundation model without ground-truth references.

**General image restoration models**  Recently, various papers have proposed image restoration models that can handle multiple tasks using the same underlying weights (Potlapalli et al., 2023; Hu et al., 2025; Cui et al., 2025). While these works share a similar goal of building foundation models, they differ from this paper in that they focus on (semi) blind image-to-image tasks, such as denoising, deraining, and dehazing. In contrast, the proposed model is designed to handle scientific and medical imaging tasks where the measurements are not necessarily in the image space and the observation model (1) is typically known, such as in MRI (k-space data) or CT (sinograms). Another recent line of work pushes beyond standard denoising priors by integrating general restoration models into iterative schemes (Hu et al., 2023; 2024; Terris et al., 2025). While these approaches use training tasks similar to ours, they rely on a model trained for a single restoration task. In contrast, our method learns a single architecture capable of handling multiple tasks in a non-iterative framework.

## 3 PROPOSED APPROACH

### 3.1 UNROLLED ARCHITECTURES

Unrolled architectures propose to "unroll" an optimization algorithm for solving equation 2, mixing interpretable blocks and learnable blocks accounting for $g$. This yields an architecture of the form

$$\mathrm{R}_\theta(\boldsymbol{y}) = L_{\theta_K} \circ D(\cdot, \boldsymbol{y}) \circ \cdots \circ L_{\theta_0} \circ D(\boldsymbol{A}^\top \boldsymbol{y}, \boldsymbol{y}), \tag{3}$$

where $(L_{\theta_k})_{0 \le k \le K}$ are learnable layers, and $D(\cdot, \boldsymbol{y})$ is a data-fidelity enforcing term, ensuring that the iterates satisfy equation 1. Two main choices for $D$ arise in this context: the first one is

$D = \mathrm{Id} - \nabla f(\cdot, \boldsymbol{y})$ ; the second one is $D = \mathrm{prox}_{\lambda f}$, where $\mathrm{prox}$ denotes the proximal operator (Combettes & Pesquet, 2011; Parikh et al., 2014), $f(\boldsymbol{x}, \boldsymbol{y}) = \frac{1}{2} \|\boldsymbol{A}\boldsymbol{x} - \boldsymbol{y}\|_2^2$ and $\lambda > 0$ is a constant. Using the proximal operator allows to accommodate choose non-differentiable data terms for $f$, which can be useful in practice. For instance, in the noiseless case, one can to choose $f$ as the indicator function of the set $\mathcal{C} = \{\boldsymbol{x} | \boldsymbol{A}\boldsymbol{x} = \boldsymbol{y}\}$. In turn, $D$ acts as a projection on $\mathcal{C}$, ensuring that $\boldsymbol{x}$ satisfies the measurement equation. In a noisy case, one can instead choose $\mathcal{C} = \{\boldsymbol{x} | \|\boldsymbol{A}\boldsymbol{x} - \boldsymbol{y}\|_2^2 \leq \varepsilon\}$, where $\varepsilon > 0$ is a user-defined constant.

From a wider perspective, these architectures benefit from an interpretable conditioning on the measurements $\boldsymbol{y}$ and the operator $\boldsymbol{A}$, without any restrictive assumption on the operator $\boldsymbol{A}$. For instance, these do not impose a matrix form for $\boldsymbol{A}$, nor an SVD decomposition. However, they strongly differ from now established architectures that have demonstrated state-of-the-art performance on low-level vision tasks (Zhang et al., 2021; Zamir et al., 2022b; Potlapalli et al., 2023). In this work, we aim at reconciling these two views, and propose to leverage tools from unrolled architectures design to condition a standard backbone model on $\boldsymbol{y}$, $\boldsymbol{A}$ and statistics of the acquisition noise.

## 3.2 PROPOSED ARCHITECTURE

The proposed architecture, illustrated in Figure 2, is derived from the DRUNet (Zhang et al., 2021). In this section, we detail the proposed architectural modifications to handle general inverse problems. First, we add a proximal estimation module casting the input into a Gaussian-noise-corrupted form. In subsequent layers, the conditioning is performed via Krylov subspace iteration on the feature maps. Finally, we adapt the architecture in order to handle a variety of noise models.

**Initialization with a proximal estimation module** In order to solve (1) for a wide variety of inverse problems, a first step is to map $\boldsymbol{y}$ onto the appropriate image domain. Several approaches are used in the literature, using either $\boldsymbol{A}^\dagger \boldsymbol{y}$ as the input of the network or $\boldsymbol{A}^\top \boldsymbol{y}$. The first approach shows the advantage of inverting the ill-conditioned $\boldsymbol{A}$, turning the effective problem for subsequent backbone layers as a Gaussian denoising problem. However, this pseudo-inverse operation is extremely sensitive to noise. In noisy settings, the latter approach is often chosen, at the cost of blurring of the input. Instead, letting $f(\boldsymbol{x}, \boldsymbol{y}) = \frac{1}{2} \|\boldsymbol{A}\boldsymbol{x} - \boldsymbol{y}\|_2^2$, our first estimation step consists in a proximal step

$$\mathrm{prox}_{\lambda f}(\boldsymbol{A}^\top \boldsymbol{y}) = \arg\min_{\boldsymbol{u}} \lambda \|\boldsymbol{A}\boldsymbol{u} - \boldsymbol{y}\|^2 + \|\boldsymbol{u} - \boldsymbol{A}^\top \boldsymbol{y}\|^2, \tag{4}$$

where $\lambda > 0$ is a regularization parameter. Equation (4) can be seen as a middle-ground between $\boldsymbol{A}^\top \boldsymbol{y}$ (when $\lambda$ is low) and $\boldsymbol{A}^\dagger \boldsymbol{y}$ (when $\lambda$ is large). We set $\lambda$ proportional to the input signal-to-noise ratio (SNR) as $\lambda = \sigma\eta/\|\boldsymbol{y}\|_1$ where $\sigma$ is the Gaussian noise level and $\eta$ is a learnable parameter.

**Krylov subspace module** Given an intermediate estimate of the image $\boldsymbol{x}^\ell$, unrolled network architectures condition on the acquisition physics $\boldsymbol{A}$ via a gradient step on an $L^2$ distance data-fidelity term, i.e.

$$\boldsymbol{x}^{\ell+1} = \boldsymbol{x}^\ell - \gamma \boldsymbol{A}^\top(\boldsymbol{A}\boldsymbol{x}^\ell - \boldsymbol{y}),$$

or via proximal step, i.e.

$$\boldsymbol{x}^{\ell+1} = \arg\min_{\boldsymbol{u}} \gamma \|\boldsymbol{A}\boldsymbol{u} - \boldsymbol{y}\|^2 + \|\boldsymbol{u} - \boldsymbol{x}^\ell\|^2.$$

for some stepsize $\gamma$. The proximal step is commonly solved using $K$ iterations of conjugate gradient, yielding a solution in the Krylov subspace spanned by $\{(\boldsymbol{I} + \gamma \boldsymbol{A}^\top \boldsymbol{A})^k (\gamma \boldsymbol{A}^\top \boldsymbol{y} + \boldsymbol{x}^\ell)\}_{k=0}^K$, and both steps can be written as

$$\boldsymbol{x}^{\ell+1} = \sum_{k=0}^{K} \alpha_k (\boldsymbol{A}^\top \boldsymbol{A})^k \boldsymbol{x}^\ell + \beta_k (\boldsymbol{A}^\top \boldsymbol{A})^k \boldsymbol{A}^\top \boldsymbol{y}, \tag{5}$$

for some scalar coefficients $\{(\alpha_k, \beta_k)\}_{k=1}^K$. Thus, we propose to condition on $\boldsymbol{A}$ using a Krylov Subspace Module (KSM), which learns the linear combination coefficients $\{(\alpha_k, \beta_k)\}_{k=1}^K$. Given an intermediate latent representation, a decoding module first maps the features onto the image domain using a convolution layer. Then, it stacks $\{((\boldsymbol{A}^\top \boldsymbol{A})^k \boldsymbol{x}^\ell, (\boldsymbol{A}^\top \boldsymbol{A})^k \boldsymbol{A}^\top \boldsymbol{y})\}_{k=1}^K$ along channels and combines them through a $3 \times 3$ convolution. The output is finally added to the latent space via a convolutional encoding module, see Figures 2 and 3 for more details.

**Multiscale operator conditioning** For a fixed number of measurements $m$, the finer the grid (i.e., larger $n$), the more ill-posed the inverse problem becomes, as the dimension of the nullspace of $\boldsymbol{A}$ is lower bounded by $n - m$. Thus, inspired by multigrid methods (Hackbusch, 2013), we propose to condition our architecture on forward operators defined on coarse grids as:

$$\boldsymbol{A}_s = \boldsymbol{A}\boldsymbol{U}_s \tag{6}$$

where $\boldsymbol{U}_s : \mathbb{R}^{\frac{n}{4^s}} \to \mathbb{R}^n$ is a $s\times$ upsampling operator with a Kaiser-windowed sinc antialias

filter (Kaiser & Schafer, 1980). Figure 3 illustrates that the linear pseudoinverse is unstable on fine grids but remains stable on coarse grids. We normalize all operators to have unit norm at each scale, i.e., $\|\boldsymbol{A}_s\|_2 = 1$ for all $s$. Moreover, for many inverse problems, we can develop efficient implementations of $\boldsymbol{A}_s^\top \boldsymbol{A}_s \in \mathbb{R}^{\frac{n}{4^s} \times \frac{n}{4^s}}$ and $\boldsymbol{A}_s^\top \boldsymbol{y} \in \mathbb{R}^{\frac{n}{4^s}}$ which can be computed fully on the coarse grid, avoiding any expensive fine scale computations. For example, if $\boldsymbol{A}$ represents a blur or an inpainting operation, we can simply downscale the kernel or the inpainting mask, respectively. We stress that, in the case where $K = 0$, the proposed method amounts to condition inner feature maps on $\boldsymbol{y}$ only. In this case, we recover the setup of (Ren et al., 2023).

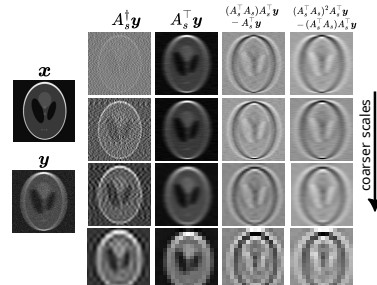

Figure 3: **Multiscale conditioning.** Estimating the underlying image (here motion blur) is easier on a coarser grid than a fine grid, as the forward operator is more ill-posed in the latter case.

**Noise conditioning & biases** Same as the original DRUNet (Zhang et al., 2018a; 2021), conditioning on the noise level is implemented by concatenating constant feature maps, filled with the noise level, along the channel dimension. While a single noise map was added for Gaussian denoising in (Zhang et al., 2021), we extend this strategy to two maps, corresponding to the parameters $\sigma$ and $\gamma$ of the Poisson–Gaussian noise model

$$\boldsymbol{y} = \gamma\boldsymbol{z} + \sigma\boldsymbol{n}, \tag{7}$$

where $\gamma \geq 0$ is the gain factor[1] associated to the Poisson noise $\boldsymbol{z} \sim \mathcal{P}(\boldsymbol{x}/\gamma)$, and $\sigma \geq 0$ is the standard deviation of the Gaussian component of the noise, with $\boldsymbol{n} \sim \mathcal{N}(\boldsymbol{0}, \boldsymbol{I})$. All biases are removed from our architecture to ensure scale equivariance with respect to both $\sigma$ and $\gamma$ components, enabling better generalization to unseen noise levels (Zhang et al., 2021; Mohan et al., 2020).

**Shared layers between imaging modalities** The modifications needed to handle inputs with different channel numbers (color, grayscale, or complex images) involve only a small subset of the model parameters. Specifically, only the first (input) and last (output) convolutional layers of the DRUNet, as well as the encoding and decoding blocks within the KSM modules, are adjusted to account for varying channel numbers. All other weights of the model are shared across modalities.

## 4 TRAINING

### 4.1 SUPERVISED TRAINING

We train our network *simultaneously* on $G$ computational imaging tasks, spanning image restoration (deblurring, inpainting, Poisson-Gaussian denoising in both color and grayscale), single-coil MRI, CT, and others (see Table 5 for a complete list), leveraging the DeepInverse library (Tachella et al., 2023a). Each task $g$ is associated with a dataset $\mathcal{D}_g = \{\boldsymbol{x}_{i,g}\}_{i=1}^{N_g}$: LSDIR (Li et al., 2023) for tasks involving natural images, LIDC-IDRI for CT images (Armato III et al., 2011), and the fastMRI brain-multicoil dataset for MRI (Zbontar et al., 2018).

Furthermore, each task $g$ can be framed as an inverse problem (1) for some $\boldsymbol{A}$, and noise following the Poisson Gaussian distribution in (7) with varying gain $\gamma \geq 0$ and standard deviation $\sigma \geq 0$. Our model is trained in a supervised fashion to minimize the following task-wise loss:

$$\mathcal{L}_g(\boldsymbol{\theta}, \boldsymbol{x}_{i,g}) = \mathbb{E}_{(\sigma_g, \gamma_g)}\mathbb{E}_{\boldsymbol{y}|\boldsymbol{x}_{i,g}} \, \omega_g \| \mathrm{R}_{\boldsymbol{\theta}}(\boldsymbol{y}, \boldsymbol{A}_g, \sigma_g, \gamma_g) - \boldsymbol{x}_{i,g}\|_1$$

---

[1] We use the convention that for $\gamma = 0$ the noise model becomes purely Gaussian.

| | | CBSD68 | | | Urban100 | | | Div2K | | | CBSD68 | | | Urban100 | | | Div2K | | |
|---|---|---|---|---|---|---|---|---|---|---|---|---|---|---|---|---|---|---|---|
| Method | Params | Easy | Med. | Hard | Easy | Med. | Hard | Easy | Med. | Hard | Easy | Med. | Hard | Easy | Med. | Hard | Easy | Med. | Hard |
| DPIR | 32M | 33.45 | 26.53 | 24.42 | **34.39** | 28.12 | 24.38 | **36.32** | 29.97 | 27.16 | 32.10 | 25.70 | 22.73 | **32.53** | 23.78 | 20.38 | **35.23** | 28.14 | 24.71 |
| DiffPIR | 552M | 31.83 | 25.98 | 23.80 | 32.93 | 27.03 | 23.69 | - | - | - | 30.93 | 24.96 | 22.51 | 31.17 | 23.63 | 20.32 | - | - | - |
| DDRM* | 552M | 33.10 | 27.97 | 25.41 | **34.19** | **29.65** | **26.05** | 35.19 | 30.57 | 27.70 | 31.85 | 25.89 | 23.14 | **32.73** | 24.40 | 20.73 | 34.46 | 28.25 | 24.89 |
| PDNet | 456K | 30.54 | 26.84 | 24.62 | 28.36 | 25.22 | 22.67 | 32.61 | 28.81 | 26.32 | 29.88 | 25.26 | 22.55 | 27.85 | 22.81 | 19.90 | 32.01 | 27.29 | 24.03 |
| Restormer | 26M | 30.96 | 26.99 | 25.00 | 28.21 | 24.75 | 23.13 | 28.56 | 25.75 | 25.00 | 31.68 | 25.93 | 23.19 | 29.07 | 24.00 | 20.99 | 30.01 | 27.20 | 25.00 |
| uDPIR/t | 32M | 33.78 | 28.20 | 25.61 | 33.07 | 28.73 | 25.42 | 35.79 | 30.76 | 27.91 | 31.90 | 26.10 | 23.41 | 30.47 | 24.31 | 21.05 | 34.63 | 28.45 | 25.20 |
| uDPIR/u | 256M | 33.64 | 27.90 | 25.54 | 32.72 | 28.14 | 25.28 | 35.38 | 30.17 | 27.77 | 31.86 | 26.03 | 23.38 | 30.26 | 24.32 | 21.04 | 34.47 | 28.37 | 25.23 |
| RAM | 36M | 34.04 | 28.22 | 25.64 | 33.61 | 28.63 | 25.30 | 36.18 | 30.72 | 27.89 | 32.59 | 26.19 | 23.42 | 31.78 | 24.65 | 21.12 | 35.23 | 28.56 | 25.16 |

Table 1: Results on motion (left) and Gaussian (right) blur. Best result in red - second best in blue. Results in bold indicate iterative methods that outperform the proposed method. *DDRM assumes access to an SVD decomposition of the operator; for fairness, we adapt the blur problem to allow FFT-based convolution.

where $\mathrm{R}_{\boldsymbol{\theta}}$ is the model to be trained, $\boldsymbol{A}_g$ is the measurement operator of task $g$, and $(\sigma, \gamma)$ are the noise parameters sampled from a distribution $p(\sigma, \gamma)$. We choose the $\ell_1$ over the standard $\ell_2$ loss as it was empirically observed to obtain better test performance (Zhao et al., 2017). We consider a weighting parameter $\omega_g = \|\boldsymbol{A}_g^\top \boldsymbol{y}\|_2 / \sigma_g$ to ensure the training loss is balanced across different noise levels and tasks. The final training loss is obtained by summing over all tasks:

$$\mathcal{L}(\boldsymbol{\theta}) = \sum_{g=1}^{G} \sum_{i=1}^{N_g} \mathcal{L}_g(\boldsymbol{\theta}, \boldsymbol{x}_{i,g}). \tag{8}$$

This formulation allows the network to generalize across multiple imaging modalities by learning from diverse measurement operators and noise distributions.

For each of the training inverse problems, we extract a random image patch $\boldsymbol{x}_{i,g}$ of size $(C, 128, 128)$ with $C \in \{1, 2, 3\}$ to which we apply the measurement $\boldsymbol{A}_g$. We use the pretrained weights of the DRUNet denoiser to initialize our model. The model is trained with a batch size of 16 per inverse problem considered and for a total of 200k steps. We use the Adam optimizer with a learning rate $10^{-4}$, which is divided by 10 after 180k steps.

## 4.2 Self-supervised finetuning

Many real computational imaging applications have limited or no ground-truth reference data (Belthangady & Royer, 2019). In such cases, the RAM model can be finetuned using measurement data alone, leveraging some of the recent advances in self-supervised learning for inverse problems (Hendriksen et al., 2020; Huang et al., 2021; Chen et al., 2021; Tachella et al., 2022; 2024). In particular, we can finetune our pretrained RAM model with a loss that only requires a dataset of noisy and/or incomplete observations $\{\boldsymbol{y}_1, \ldots, \boldsymbol{y}_N\}$:

$$\mathcal{L}(\boldsymbol{\theta}) = \sum_{i=1}^{N} \mathcal{L}_{\mathrm{MC}}(\boldsymbol{\theta}, \boldsymbol{y}_i) + \omega \, \mathcal{L}_{\mathrm{NULL}}(\boldsymbol{\theta}, \boldsymbol{y}_i), \tag{9}$$

where $\omega > 0$ is a trade-off parameter. The first term, $\mathcal{L}_{\mathrm{MC}}$, enforces measurement consistency $\boldsymbol{A}\hat{\boldsymbol{x}} \approx \boldsymbol{A}\boldsymbol{x}$ while taking care of the noise, and is chosen according to the knowledge about the noise distribution of the finetuning dataset, i.e., using SURE (Stein, 1981) for fully-specified noise models, and UNSURE (Tachella et al., 2024) or splitting losses (Batson & Royer, 2019) in the case of unknown or mispecified noise models. The second term, $\mathcal{L}_{\mathrm{NULL}}$, is required if $\boldsymbol{A}$ is non-invertible and handles the lack of information in the nullspace of $\boldsymbol{A}$, by leveraging information from multiple forward operators (Tachella et al., 2023b) and/or enforcing equivariance to a group of transformations (Chen et al., 2021). More details are provided in Appendix E.1.

## 5 Results

### 5.1 Baselines

We evaluate our model's performance alongside both iterative reconstruction methods, unrollled models and end-to-end architectures. First, we consider the Plug-and-Play model DPIR (Zhang

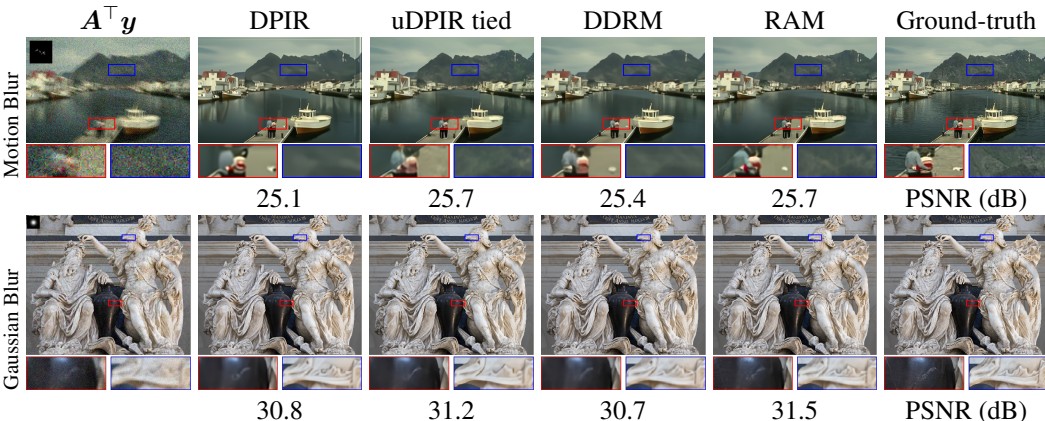

Figure 4: **Deblurring results.** Top row: motion blur hard, on a CBSD68 sample. Bottom row: Gaussian blur medium, on a DIV2K sample.

| Method | MRI ×4 | | MRI ×8 | | CT | | SR4 | |
|---|---|---|---|---|---|---|---|---|
| | PSNR | SSIM | PSNR | SSIM | PSNR | SSIM | Clean | Noisy |
| PDNet | 28.25 | 0.719 | 24.54 | 0.641 | 23.09 | 0.713 | - | - |
| DPIR | 30.54 | 0.784 | 25.28 | 0.661 | - | - | 25.67 | 25.59 |
| uDPIR-u | 33.73 | 0.848 | 30.20 | 0.792 | 27.00 | 0.772 | - | - |
| uDPIR-t | 34.14 | 0.851 | 30.86 | 0.805 | 28.35 | 0.779 | - | - |
| SWINIR | - | - | - | - | - | - | 26.16 | 25.47 |
| RAM | 34.39 | 0.853 | 31.50 | 0.813 | 28.83 | 0.798 | 26.04 | 25.47 |

| Method | mcMRI ×8 | | Poisson CT | |
|---|---|---|---|---|
| | PSNR | SSIM | PSNR | SSIM |
| PDNet | - | - | 13.20 | 0.357 |
| UNet | 28.50 | 0.662 | - | - |
| uDPIR-untied | 33.71 | 0.850 | 14.62 | 0.446 |
| uDPIR-tied | 36.06 | 0.894 | 14.67 | 0.462 |
| RAM | 35.62 | 0.889 | 28.83 | 0.798 |

Table 2: **Left: In-distribution tasks**. MRI tasks are evaluated on the fastMRI validation set, CT tasks on the LIDC-IDRI test set, and SR4 on CBSD68. **Right: Out-of-distribution tasks**, namely multi-coil MRI with acceleration factor 8 on fastMRI validation set, and computed tomography with Poisson noise on LIDC-IDRI test set.

et al., 2020), which plugs a DRUNet, trained only for denoising, within an 8-step Half Quadratic Splitting (HQS) algorithm (Aggarwal et al., 2019), thereby requiring approximately $8\times$ as many FLOPs and being $3.7\times$ slower at inference in comparison to our model (see Tables 8 and 9 for details). We also compare with unrolled versions of DPIR (referred to as uDPIR (Zhang et al., 2020)), also comprising eight HQS iterations, that we train on the same tasks, with the same loss function and dataset as our model. Specifically, we investigate both the weight-tied variant, where all eight iterations share the same DRUNet parameters, and the weight-untied variant, where each iteration has its own set of parameters—resulting in a model with eight times more parameters and FLOPs than RAM. Additionally, we compare our approach with the PDNet unrolled network (Adler & Öktem, 2018), trained on the same tasks. Finally, we evaluate Restormer (Zamir et al., 2022a), a transformer-based model with a similar computational cost (in terms of FLOPs) to DRUNet, but trained separately on each restoration tasks. We also compare the proposed model with the diffusion-based DDRM algorithm (Kawar et al., 2022) with the diffusion model backbone from (Dhariwal & Nichol, 2021); see Appendix D for technical details on adaptation of DDRM to the proposed setting.

## 5.2 IN-DISTRIBUTION RESULTS

We first evaluate the proposed method on tasks that are consistent with the training tasks.

**Deblurring** We evaluate two types of deblurring: Gaussian deblurring, using fixed Gaussian blur kernels, and motion deblurring, based on random motion blur kernels. For each, we define three difficulty levels (easy, medium, hard), determined by kernel characteristics and the Gaussian noise standard deviation. Details for each setup are provided in Appendix B.2. Figure 4 and Table 8 report visual and quantitative results. Among non-iterative methods, RAM and uDPIR-tied achieve the best performance, while uDPIR-untied underperforms despite its larger parameter count. Our method generally surpasses other end-to-end approaches, except for motion deblurring on Urban100. Compared to iterative methods, our approach is superior except at low noise levels ($\sigma = 0.01$) on large images (Urban100, DIV2K).

**MRI** We evaluate on the fastMRI brain validation set using the single-coil accelerated acquisition procedure from Zbontar et al. (2018) with acceleration factors 4 and 8. The problem is formulated as $\boldsymbol{y} = \mathrm{diag}(\boldsymbol{m})\boldsymbol{F}\boldsymbol{x} + \sigma\boldsymbol{n}$, where $\boldsymbol{m}$ is a binary mask and $\boldsymbol{F}$ the discrete Fourier transform, and $\sigma = 5 \cdot 10^{-4}$. Quantitative results are reported in Table 2 and visuals in Figure 9. The proposed RAM model outperforms all baselines.

**CT** We evaluate on computed tomography with Gaussian noise, modeled as $\boldsymbol{y} = \boldsymbol{A}\boldsymbol{x} + \sigma\boldsymbol{n}$, where $\boldsymbol{A}$ is the Radon transform. The acquisition uses 51 angles, i.e., about $10\%$ of the LIDC-IDRI image width, with $\sigma = 10^{-4}$ as in training. Results are reported in the rightmost column of Table 2 and visuals in Figure 10. DPIR results are omitted due to instability on this task. The proposed RAM model produces reconstructions with finer details than uDPIR-tied.

Additional results for in-distribution tasks on Poisson-Gaussian denoising, motion blur, super-resolution, and inpainting are provided in Appendix G.

### 5.3 ZERO-SHOT PERFORMANCE ON OUT-OF-DISTRIBUTION TASKS

**Multi-coil MRI** We consider the Cartesian multi-coil MRI problem. In this setup, and assuming $L$ coils, the signal measured by each coil $\ell \in \{1, \cdots, L\}$ writes as $\boldsymbol{y}_\ell = \mathrm{diag}(\boldsymbol{m})\boldsymbol{F}\mathrm{diag}(\boldsymbol{s}_\ell)\boldsymbol{x} + \sigma\boldsymbol{n}_\ell$, where the $(\boldsymbol{s}_\ell)_{1 \leq \ell \leq L}$ are the sensitivity maps (or S-maps). We provide reconstruction metrics for the multi-coil setting in Table 2 and associated visuals in Figure 10. We provide comparisons with the baseline UNet from Zbontar et al. (2018). Both methods perform similarly up to the addition of mild residual noise with the UNet, yielding lower PSNR despite similar visual results. We stress that since PDNet contains learnable layers acting in the measurement domain, one would need to retrain the architecture for this new setting specifically, hence we do not present the results.

**Computed Tomography with Poisson noise** While our model was trained on the CT problem with Gaussian noise, we propose to apply it to a CT problem degraded with Poisson noise only. As previously, we consider a problem with 51 angles. Numerical results are provided in Table 2 and visual results in Figure 10. The proposed method performs better at estimating the textures.

Additional results for out-of-distribution tasks are provided in Appendix G.

### 5.4 ZERO-SHOT PERFORMANCE ON BLIND AND NON-LINEAR INVERSE PROBLEMS

Despite being trained to solve non-blind linear inverse problems, the proposed model can be easily extended beyond this setting.

**Blind deblurring** The proposed model can be used for solving blind deblurring problems, where the forward operator $\boldsymbol{A}$ is an unknown space-varying kernel, by leveraging a blur prediction network (Carbajal et al., 2023) that provides an estimate $\hat{\boldsymbol{A}}$. Figure 5 shows blind deblurring results on a real motion blur dataset (Köhler et al., 2012) using the spatially-varying blur estimator proposed by Carbajal et al. (2023). Despite being trained on exact forward operators, RAM obtains good results using inexact estimates of the operator. Additional results are presented in Appendix G.

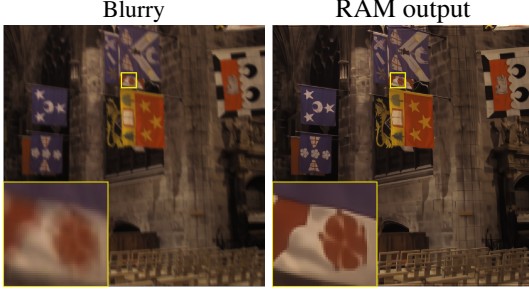

Blurry        RAM output

Figure 5: **Blind deblurring.** Results on a real motion blur (Kohler dataset).

**Phase retrieval** Although the model is only trained to solve linear inverse problems, it can still be applied to non-linear problems that can be decomposed into a sequence of linear problems. We illustrate this idea with the phase retrieval problem $\boldsymbol{y} = \boldsymbol{A}(\boldsymbol{x}) = |\boldsymbol{B}\boldsymbol{x}|$ with random matrix $\boldsymbol{B} \in \mathbb{C}^{m \times n}$ and complex $\boldsymbol{x} \in \mathbb{C}^n$. Inspired by the Gerchberg–Saxton algorithm (Gerchberg, 1972), we can iterate between updating the estimate of $\boldsymbol{x}$ given an estimate of the phase of $\boldsymbol{y}$, and updating the estimation of the phase given $\boldsymbol{x}$ (see Algorithm 2). Appendix I presents results varying the number of measurements, showing that RAM can obtain stable estimates for low $m/n$ ratios, where standard methods such as gradient descent fail.

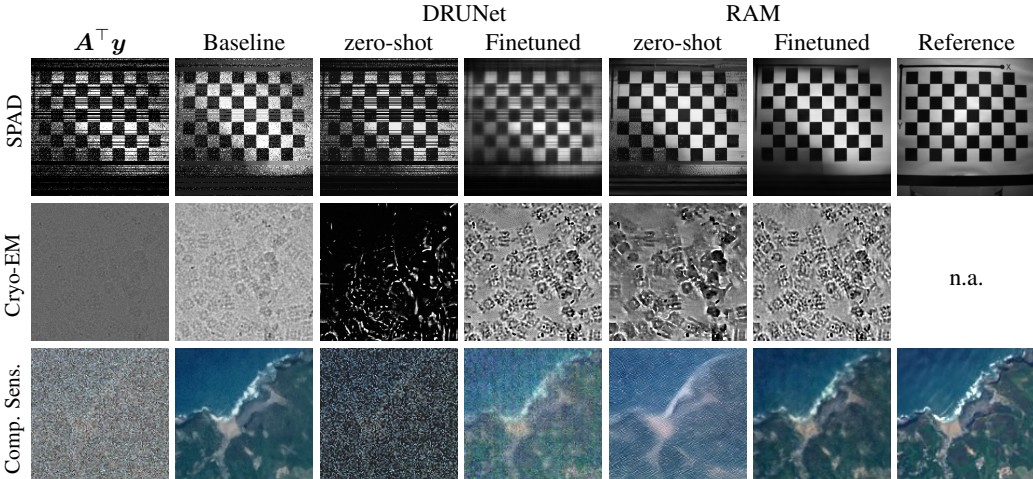

Figure 6: **Self-supervised finetuning.** RAM can be finetuned on a few real LinoSPAD or Cryo-EM images without ground-truth nor full knowledge of the noise distribution. Adaptation to new problems (compressed sensing and demosaicing) and new distributions (Sentinel 2 images), by fine-tuning on measurements of a single image.

| | | Compressed Sensing | | | | | | Demosaicing | | | | |
|---|---|---|---|---|---|---|---|---|---|---|---|---|
| | | DRUNet rand | | DRUNet | | RAM | | | | DRUNet rand | | DRUNet | | RAM | |
| Model | DPIR+ | Sup. | Self. | Sup. | Self. | Sup. | Self. | DPIR+ | DDRM | Sup. | Self. | Sup. | Self. | Sup. | Self. |
| zero-shot | 31.29 | 8.07 | | 10.23 | | 22.50 | | 32.84 | 30.64 | 8.10 | | 10.73 | | 29.49 | |
| $N=1$ | | 20.92 | 21.20 | 19.49 | 19.44 | 30.77 | 30.40 | | | 23.14 | 21.28 | 25.82 | 25.57 | 34.46 | 33.89 |
| $N=10$ | | 22.59 | 22.00 | 25.99 | 24.74 | 32.52 | 32.29 | | | 24.85 | 21.85 | 33.14 | 31.38 | 35.03 | 34.73 |
| $N=100$ | | 22.73 | 22.24 | 28.38 | 30.58 | 33.40 | 33.57 | | | 25.84 | 23.22 | 34.06 | 34.56 | 35.26 | 35.10 |

Table 3: **Effect of finetuning dataset size and self-supervision**. RAM requires a few observations to obtain good results, whereas finetuning a DRUNet denoiser requires significantly larger datasets. We report average test PSNR in dB.

## 5.5 SELF-SUPERVISED FINETUNING ON OUT-OF-DISTRIBUTION TASKS

Real imaging problems may involve inverse problems or images that significantly differ from those at train time. In this case, finetuning with only a handful of measurements (i.e., without ground-truth data), sometimes a single one, substantially improves performance. We present results for three different tasks in Figure 6 and Table 3. In all cases, finetuning with fewer than $N = 10$ measurements results in performances that significantly outperform PnP and diffusion baselines. Moreover, finetuning takes a few minutes on a single consumer-grade GPU (see Table 7).

**Compressed sensing on Sentinel-2 data:** on 100 Sentinel-2 RGB images ($128 \times 128$, 10m resolution), we set $A = S\text{diag}(m)$ with $m$ a random $\pm 1$ mask and $S$ a subsampled sine transform (factor 4). With Gaussian noise $\sigma = 0.05$, RAM finetuning achieves strong results from a single image, clearly surpassing DRUNet. **Cryo electron microscopy:** we use 5 noisy Cryo-EM micrographs ($7676 \times 7420$) from Topaz-EM (Bepler et al., 2020), where the SNR is very low. Since the noise distribution is unknown, we finetune with a splitting loss. **Low-photon imaging:** we use 9 LinoSPAD images (Lindell et al., 2018) ($256 \times 256$). While noise is generally assumed to be Poisson in the low-flux case, the images were acquired on high-flux conditions and follow an unknown discrete noise distribution. Due to missing pixel lines, recovery amounts to inpainting and denoising. We finetune with a splitting and multi-operator losses (removing lines at random). Further implementation details are provided in Appendix E.2.

## 6 ABLATIONS, UNCERTAINTY QUANTIFICATION, AND LIMITATIONS

**Architectural ablations** Table 4 shows training PSNRs obtained on all training tasks after a fixed budget of $10^4$ iterations for various architectural variations. We first note that introducing a proximal input block leads to a substantial improvement (+0.8dB). Conditioning the inner feature maps on the observation $y$ provides a smaller but consistent improvement (+0.1dB), an observation in line with (Ren et al., 2023). The largest performance boost (+0.9dB) comes from incorporating our Krylov blocks, which apply $A^\top A$ operations to the feature maps. This highlights that mimick-

| Configuration | PSNR (dB) | #Params | | Training tasks | Inpainting | Deblurring | SR ($\times$2) |
|---|---|---|---|---|---|---|---|
| base (DRUNet backbone) | 25.83 | 32.6M | | Inpainting only | 30.84 | 23.03 | 26.17 |
| base + prox | 26.64 | 32.6M | | Deblurring only | 14.06 | 25.62 | 28.02 |
| base + prox + embed $y$ | 26.71 | 33.4M | | SR$\times$2, $\sigma = 0$ only | n/a | 21.25 | 28.60 |
| base + prox + embed $y$ + Krylov (RAM) | 27.61 | 35.6M | | All three tasks | 30.73 | 25.58 | 29.79 |

Table 4: **Ablations**. Left: Architectural ablations. Average training PSNR on the training set for different architecture variants. Right: Training task ablation. Performance of models trained on different task subsets on the validation set.

ing the core operations of iterative optimization methods within the architecture yields significant benefits. Interestingly, our model surpasses unrolled architectures that are explicitly derived from optimization algorithms. Comparisons with a PromptIR backbone are provided in Appendix C.

**Single vs multitask training** Table 4 shows that given appropriate conditioning on the measurement operator, bias towards specific training tasks is negligible, as the model trained on all tasks shows a performance comparable (yet slightly inferior to) models trained on specific tasks only. The proposed reweighting using the SNR of the considered problem limits bias towards harder tasks.

**Uncertainty quantification** We can use existing uncertainty quantification (UQ) algorithms that only require a general reconstruction function. Figure 7 shows estimated errors using the recent equivariant bootstrap technique (Tachella & Pereyra, 2024) (see Algorithm 1 on a noisy image inpainting task (DIV2K validation images with $\sigma = 0.02$ and 50% observed pixels). More details about the bootstrapping algorithm and additional results are included in Appendix H. As shown in Figure 19, our

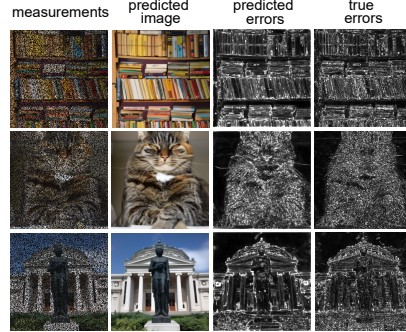

Figure 7: **Uncertainty quantification.** Equivariant bootstrapping with RAM provides estimates of pixelwise errors that are close to the true errors.

algorithm obtains better-calibrated uncertainty estimates than DDRM, while requiring $100\times$ fewer network evaluations.

**Limitations** Training the RAM model requires a fair amount of GPU resources, which might not be available to all practitioners. Nevertheless, our model can be finetuned on a single mid-sized GPU, obtaining competitive results with a few optimization steps on small finetuning datasets. We focus on low-distortion reconstructions, in contrast to the higher perceptual quality and higher distortion of diffusion methods (Blau & Michaeli, 2018). However, reconstructors such as RAM can be used within sampling algorithms, e.g. (Delbracio & Milanfar, 2023; Ohayon et al., 2025).

# 7 CONCLUSION

We present a new lightweight foundational model for computational imaging that achieves state-of-the-art performance on a wide range of problems, outperforming PnP methods while providing similar results to $8\times$ more compute and parameter-intensive unrolled networks. Our results demonstrate competitive performances without following the common practice of unrolling an optimization algorithm, resulting in a faster and lighter reconstruction network.

Our model displays transfer capabilities, as it can be finetuned on new datasets or imaging problems with small datasets (up to a single image) in a fully self-supervised way, i.e., without any ground-truth references. These results challenge the idea that a reconstruction network has to be specialized to a specific imaging task, showing good reconstructions across a wide variety of imaging modalities with a relatively lightweight network (36M parameters). Our results suggest that imaging tasks that might seem very different (e.g. cryo-EM denoising and demosaicing of satellite images) share significant common structure and can be tackled with the same model.

We believe this work paves the way for a new approach to imaging, where most of the effort can be put into developing strong base models and robust self-supervised finetuning techniques.

ACKNOWLEDGEMENTS

M. Terris acknowledges support from the BrAIN grant (ANR-20-CHIA-0016). M. Song acknowledges support from the PNRIA CNRS project. J. Tachella acknowledges support from the ANR grant UNLIP 23-CE23-0013. This work used HPC resources from GENCI–IDRIS (Grants 2024-AD011014344R1, 2025-AD011015191R1, 2025-AD011016833).

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

| | Task | Dataset | Configuration | Noise Distribution | | | |
| | | | | Gaussian | | Poisson | |
| | | | | $\sigma_{\min}$ | $\sigma_{\max}$ | $\gamma_{\min}$ | $\gamma_{\max}$ |
|---|---|---|---|---|---|---|---|
| Grayscale | Denoising | LSDIR | - | 0.001 | 0.2 | 0.01 | 1.0 |
| | Deblurring | LSDIR | Random motion & Gaussian kernels (31×31) | 0.001 | 0.2 | - | - |
| | Inpainting | LSDIR | Bernoulli masks: $p \sim \mathcal{U}(0.3, 0.9)$ | 0.01 | 0.2 | 0.01 | 1.0 |
| | SR×2,4 | LSDIR | Bicubic/bilinear downsampling (×2, ×4) | 0.001 | 0.01 | - | - |
| | Tomography | LIDC-IDRI | Radon transform, 10 projection angles | - | 0.01 | - | - |
| Color | Denoising | LSDIR | - | 0.001 | 0.2 | 0.01 | 1.0 |
| | Deblurring | LSDIR | Random motion & Gaussian kernels (31×31) | 0.01 | 0.2 | - | - |
| | Inpainting | LSDIR | Bernoulli masks: $p \sim \mathcal{U}(0.3, 0.9)$ | 0.01 | 0.2 | 0.01 | 1.0 |
| | SR×2,4 | LSDIR | Bicubic/bilinear downsampling (×2, ×4) | 0.001 | 0.01 | - | - |
| | Pan-sharpening | LSDIR | Flat spectral response | 0.01 | 0.1 | - | - |
| Complex | MRI | FastMRI | Acceleration masks (×4, ×8 undersampling) | $0.001^{*}$ | $0.1^{*}$ | - | - |
| | Denoising | FastMRI | - | $0.001^{*}$ | $0.1^{*}$ | - | - |
| | Inpainting | FastMRI | Bernoulli masks: $p \sim \mathcal{U}(0.1, 0.9)$ | $0.01^{*}$ | $0.1^{*}$ | - | - |

Table 5: Summary of training datasets and inverse problems. Noise parameters: when both $\sigma_{\min}$ and $\sigma_{\max}$ are specified, $\sigma_g \sim \mathcal{U}(\sigma_{\min}, \sigma_{\max})$; when only $\sigma_{\max}$ is given, $\sigma_g = \sigma_{\max}$. Same notation applies to Poisson noise parameters $\gamma_g$. $^{*}$for fastMRI, the effective noise levels need to be rescaled by a factor $5 \cdot 10^{-3}$ due to a rescaling of the full dataset. "-" indicates no noise of that type.

## A    PRE-PROCESSING OF TRAINING DATASETS

**LSDIR**    We use the LSDIR dataset (Li et al., 2023) for training on natural imaging tasks, consisting of 84,991 high quality images. We split the images in 512×512 non-overlapping patches for faster data loading. Beyond this, no additional pre-processing is performed on the dataset.

**MRI**    We perform virtual coil-combination on the raw kspace data from the brain-multicoil fastMRI dataset (Zbontar et al., 2018). We use the resulting 70,748 complex images from the training set for training our model and the 21,842 slices from the validation set are reserved for validation.

**LIDC-IDRI**    We use the LIDC-IDRI dataset (Armato III et al., 2011) as a basis for our computed tomography experiments containing 244,526 chest slices. We normalize the scans in Houndsfield Units (HU) using the rescale slope and intercept provided in the Dicom files. Data is then clipped in the lung window values [-1200, 800] and the rescaled in [0, 1]. We perform random extraction of slices from the dataset to provide a (95%, 4%, 1%) split of (training, validation, test).

## B    CONSIDERED INVERSE PROBLEMS

### B.1    TRAINING INVERSE PROBLEMS

We provide in Table 5 a detailed list of all datatets and inverse problems considered for the training of RAM.

### B.2    BLUR TASKS DEFINITION

The deblurring inverse problem is $\boldsymbol{y} = \boldsymbol{k} * \boldsymbol{x} + \sigma \boldsymbol{n}$, where $\boldsymbol{k}$ is some blur kernel and $\boldsymbol{n} \sim \mathcal{N}(\boldsymbol{0}, I)$. We use a "valid" padding strategy, i.e., the image is not padded before convolving with the blur kernel. In each case, we establish 3 types of problems (easy, medium and hard) corresponding to different kernel lengths and noise standard deviations.

**Motion deblurring**    We generate random motion blur kernels as random Gaussian processes following (Tachella et al., 2023a; Schuler et al., 2015) of $31 \times 31 pixels$. In this case, the difficulty of

the deblurring problem is defined by the length scale of blur trajectories $\ell$, the standard deviation of the Gaussian processes $s$ generating the psf and the standard deviation $\sigma$ of the additive Gaussian noise. The tuple $\{\ell, s, \sigma\}$ are set to $\{0.1, 0.1, 0.01\}$, $\{0.6, 0.5, 0.05\}$, $\{1.2, 1.0, 0.1\}$ for the easy, medium and hard settings respectively.

**Gaussian deblurring**  We generate fixed blur kernels with psf of size 31; the difficulty of the problem is defined by the standard deviation of the (Gaussian) blur kernel $\sigma_{\text{blur}}$ and the standard deviation $\sigma$ of the additive Gaussian noise. The tuple $\{\sigma_{\text{blur}}, \sigma\}$ is set to $\{1.0, 0.01\}$, $\{2.0, 0.05\}$, $\{4.0, 0.1\}$ for the easy, medium and hard settings respectively.

**Blind deblurring**  In the blind deblurring experiments, we consider the spatially-varying blurs of Carbajal et al. (2023), defined as

$$\boldsymbol{Ax} = \sum_{i=1}^{25} \boldsymbol{k}_i * (\boldsymbol{\omega}_i \circ \boldsymbol{x})$$

where $\{\boldsymbol{k}_i\}_{i=1}^{25}$ is a set of blur kernels of $33 \times 33$ pixels, and $\{\boldsymbol{\omega}_i\}_{i=1}^{25}$ is a set of spatial masks with values between 0 and 1 matching the size of the input image.

## C    COMPARISON WITH PROMPTIR BACKBONE

In the setting of Section 6 with fixed training budget, we provide in Table 6 PSNR metric for Motion deblurring on CBSD68. The DRUNet backbone performs on par with PromptIR. A significant improvement is brought by the proposed conditioning. Associated visual results are provided in Figure 8 in the $\sigma = 0.05$ case. We notice that beyond border effects, no significant visual difference is noticed between the PromptIR and DRUNet backbones: both reconstructions show hallucinated archs that are present in the measurement but not in the groundtruth. The proposed conditioning resolves this issue.

|  | PromptIR | DRUNet | RAM |
|---|---|---|---|
| Motion blur $\sigma = 0.01$ | 22.84 | 22.99 | 30.78 |
| Motion blur $\sigma = 0.05$ | 22.42 | 22.46 | 26.47 |
| Motion blur $\sigma = 0.1$ | 21.98 | 22.05 | 24.73 |

Table 6: **PSNR on the BSD68 dataset** for different architectures. The three architectures are trained in identical setups.

## D    ADAPTATION OF DDRM

The DDRM algorithm builds on a denoising diffusion UNet backbone, following the architecture from (Dhariwal & Nichol, 2021), as in (Kawar et al., 2022). We use the pretrained weights provided by the authors of (Kawar et al., 2022). Due to its large receptive field, this convolutional UNet requires sufficiently large input sizes, but memory constraints make resolutions beyond $512{\times}512$ impractical. To address this, we use circular padding to align inputs with the $2^5$ minimum size constraint, and process larger images by dividing them into non-overlapping $512{\times}512$ patches.

## E    SELF-SUPERVISED FINETUNING

### E.1    SELF-SUPERVISED LOSSES

In this section, we review the possible choices for $\mathcal{L}_{\text{MC}}$ and $\mathcal{L}_{\text{NULL}}$ that can be used in (9).

**Measurement consistency**  $\mathcal{L}_{\text{MC}}$ can be selected based on prior knowledge of the noise distribution (Tachella et al., 2024). If the noise distribution is known exactly, Stein's Unbiased Risk Estimate

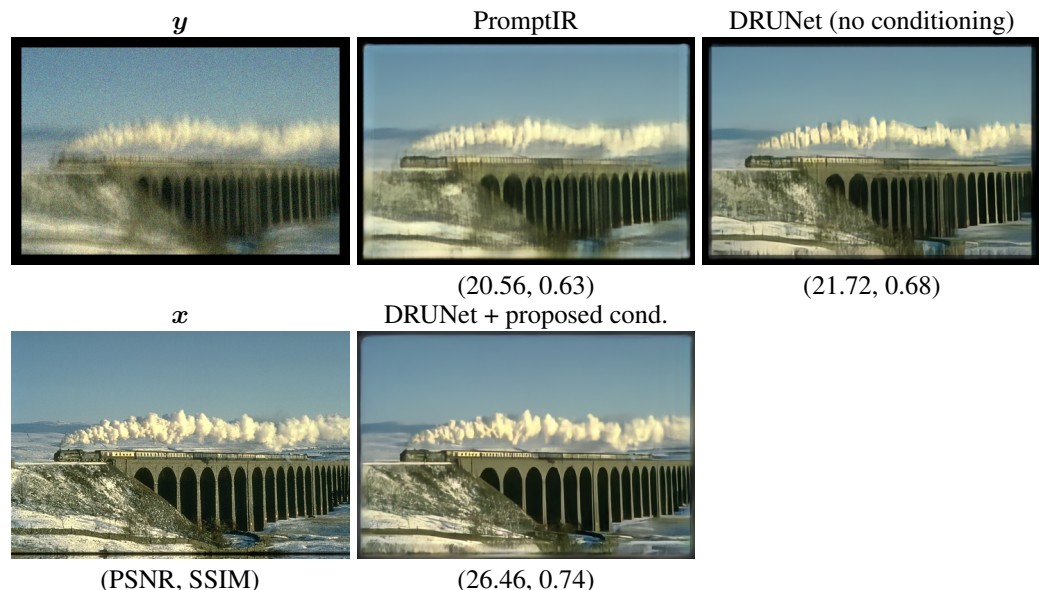

Figure 8: **Backbone & conditioning influence.** Reconstruction results on a motion blur measurement $y$ (top left) from the BSD68 dataset. Bottom center image shows the proposed RAM architecture (DRUNet backbone with proposed conditioning).

(SURE) (Stein, 1981) is used. For the case of Gaussian noise of level $\sigma$, SURE is defined[2] as

$$\mathcal{L}_{\text{SURE}}(\boldsymbol{\theta}, \boldsymbol{y}_i) = \|\boldsymbol{A}_i\, \text{R}_{\boldsymbol{\theta}}(\boldsymbol{y}_i, \boldsymbol{A}_i) - \boldsymbol{y}_i\|_2^2 + 2\sigma_i^2 \text{div}(\boldsymbol{A}_i \circ \text{R}_{\boldsymbol{\theta}})(\boldsymbol{y}_i, \boldsymbol{A}_i),$$

where the divergence is approximated using a Monte Carlo method (Ramani et al., 2008). Extensions of SURE to unknown $\sigma$ (Tachella et al., 2024) and other noise distributions also exist (Monroy et al., 2025). If the noise distribution is only assumed to be separable across measurements, we can use a splitting loss

$$\mathcal{L}_{\text{SPLIT}}(\boldsymbol{\theta}, \boldsymbol{y}_i) = \mathbb{E}_{\boldsymbol{m}} \|(\boldsymbol{I} - \text{diag}(\boldsymbol{m}))\, (\boldsymbol{A}_i\, \text{R}_{\boldsymbol{\theta}}(\text{diag}(\boldsymbol{m})\boldsymbol{y}_i, \text{diag}(\boldsymbol{m})\boldsymbol{A}_i) - \boldsymbol{y}_i)\|_2^2,$$

where $\boldsymbol{m}$ is a splitting mask sampled from a Bernoulli random variable or using problem-specific strategies (e.g., Neighbor2Neighbor (Huang et al., 2021), SSDU (Yaman et al., 2020), etc.).

**Learning in the nullspace**    If the finetuning dataset is observed via a single operator $\boldsymbol{A}_i = \boldsymbol{A}$ for all $i = 1, \dots, N$, we choose $\mathcal{L}_{\text{NULL}}$ as the Equivariant Imaging (EI) loss (Chen et al., 2021) which enforces equivariance to a group of transformations $\{\boldsymbol{T}_r\}_{r \in \mathcal{R}}$ such as rotations or shifts with

$$\mathcal{L}_{\text{EI}}(\boldsymbol{\theta}) = \mathbb{E}_{r \sim \mathcal{R}} \|\boldsymbol{T}_r \hat{\boldsymbol{x}}_{i,\boldsymbol{\theta}} - \text{R}_{\boldsymbol{\theta}}(\boldsymbol{A}\boldsymbol{T}_r \hat{\boldsymbol{x}}_{i,\boldsymbol{\theta}}, \boldsymbol{A})\|_2^2,$$

where $\hat{\boldsymbol{x}}_{i,\boldsymbol{\theta}} = \text{R}_{\boldsymbol{\theta}}(\boldsymbol{y}_i, \boldsymbol{A}_i)$ is the reconstructed image. To learn in the nullspace of $\boldsymbol{A}$, the group of transformations should be chosen such that $\boldsymbol{A}$ is not equivariant. If the finetuning dataset is associated with multiple operators $\{\boldsymbol{A}_r\}_{r \in \mathcal{R}}$, we choose $\mathcal{L}_{\text{NULL}}$ as the Multi Operator Imaging (MOI) loss (Tachella et al., 2022), i.e.,

$$\mathcal{L}_{\text{MOI}}(\boldsymbol{\theta}) = \mathbb{E}_{r \sim \mathcal{R}} \|\hat{\boldsymbol{x}}_{i,\boldsymbol{\theta}} - \text{R}_{\boldsymbol{\theta}}(\boldsymbol{A}_r \hat{\boldsymbol{x}}_{i,\boldsymbol{\theta}}, \boldsymbol{A}_r)\|_2^2,$$

also where $\hat{\boldsymbol{x}}_{i,\boldsymbol{\theta}} = \text{R}_{\boldsymbol{\theta}}(\boldsymbol{y}_i, \boldsymbol{A}_i)$, which enforces consistency across operators $\text{R}_{\boldsymbol{\theta}}(\boldsymbol{A}_r \boldsymbol{x}, \boldsymbol{A}_r) \approx \text{R}_{\boldsymbol{\theta}}(\boldsymbol{A}_s \boldsymbol{x}, \boldsymbol{A}_s)$ for all pairs $(i, r)$.

### E.2    EXPERIMENTAL DETAILS

In all finetuning experiments, we use the Adam optimizer with the same parameters used at train time (see Section 4.1). In all experiments and baselines, we choose the network checkpoint that

---

[2]We do not explicit the dependency of $\text{R}_{\boldsymbol{\theta}}$ on $(\sigma, \gamma)$.

obtains the best performance, using ground-truth for the Sentinel experiments and visual inspection for the SPAD and Cryo-EM data. The trade-off parameter in (9) is set as $\omega = 0.1$ in all experiments where a $\mathcal{L}_{\text{NULL}}$ loss is used. Shifts up to $10\%$ of the image width and height are considered in the EI loss. Figure 6 shows reconstructions with finetuning on a single measurement on a compressed sensing problem, and Figures 16 and 17 show more reconstructions for the LinoSPAD and Cryo-EM datasets, respectively.

Table 7: **Self-supervised finetuning timings.** Experiments on Sentinel 2 data (Tab. 5 of main), using a single RTX 4090 GPU.

| Dataset | Compressed Sensing | Demosaicing |
|---|---|---|
| 1 image | 43 sec | 60 sec |
| 10 images | 80 sec | 107 sec |
| 100 images | 228 sec | 267 sec |

**Satellite images**  We use a dataset of 200 images of $128 \times 128$ pixels of the coast of the United Kingdom taken by the Sentinel-2 L1C satellite with 10-meter resolution and minimal cloud coverage, keeping only the RGB bands. The dataset is split into 100 images for training and 100 for testing.

- **Compressed sensing**: we set $A = S\text{diag}(m)$ where $m$ is a random mask with values in $\{-1,+1\}$, and $S \in \mathbb{R}^{m \times n}$ is the discrete sine transform wit output randomly subsampled by a factor of 4.

- **Demosaicing**: measurements are generated by applying a Bayer pattern, keeping a single band per pixel. In both cases, we consider Gaussian noise with $\sigma = 0.05$, and finetune with the $\ell_2$ loss for the supervised case, and with $\mathcal{L}_{\text{SURE}}$ and $\mathcal{L}_{\text{EI}}$ (using shifts) for the self-supervised setting. As shown in Table 3 and Figure 6, finetuning RAM requires only on *a single image* to obtain good results, significantly outperforming the DRUNet baselines. As shown in Table 7, self-supervised finetuning can be performed in the order of a couple of minutes on a single mid-sized GPU.

**Cryo electron microscopy**  We evaluate the model on 5 real Cryo-EM images of $7676 \times 7420$ pixels provided by the Topaz-EM open-source library (Bepler et al., 2020) whose noise distribution is unknown and have very low SNR. We finetune our model with fixed $A = I$, $\sigma = 0.98$, $\gamma = 0$ at the input, using $256 \times 256$ crops normalized to have unitary standard deviation. Since the noise distribution is unknown and there is a trivial nullspace, we use $\mathcal{L}_{\text{SPLIT}}$ for measurement consistency and $\mathcal{L}_{\text{NULL}} = 0$. Reconstructions are shown in Figure 6.

**Low-photon imaging**  We evaluate the RAM model on the LinoSPAD data provided by Lindell et al. (2018), which is corrupted by photon noise. While noise in SPADs is generally assumed to be Poisson in the very low-flux case, images acquired with higher flux can follow more complex discrete noise distributions. The LinoSPAD scans the image using epipolar scanning, with some lines of pixels removed due to faulty acquisition. Thus, the image recovery problem in this case is inpainting. The missing line pattern and noise model were not used during model training. We finetune the model using $\mathcal{L}_{\text{SPLIT}}$ and $\mathcal{L}_{\text{MOI}}$ (removing random subsets of lines). Reconstructions are shown in Figure 6.

| | PDNet | uDPIR-untied | uDPIR-tied | RAM |
|---|---|---|---|---|
| GFLOPS | 60 | 2234 | 2234 | 360 |
| Test memory (BS=1) | 52 | 1213 | 298 | 354 |
| Train memory (BS=8) | 5815 | 37288 | 36374 | 9670 |

Table 8: **Computational metrics.** Floating point operations (FLOPs) and memory requirements in MBytes for different methods on $256 \times 256$ color image motion deblurring.

|  | DDRM | uDPIR tied | RAM (ours) |
|---|---|---|---|
| Denoising $A = I$ | $5746.83 \pm 3.59$ | $177.08 \pm 0.23$ | $\mathbf{45.21 \pm 0.13}$ |
| Inpainting/masking $A = \text{diag}(m)$ | $5759.69 \pm 12.82$ | $177.95 \pm 0.15$ | $\mathbf{45.78 \pm 0.08}$ |
| Blur $A = F\text{diag}(k)F^{-1}$ | $5773.47 \pm 2.77$ | $178.54 \pm 0.21$ | $\mathbf{48.24 \pm 0.06}$ |
| Generic $A = \text{diag}(m)F\text{diag}(k)F^{-1}$ | Not available | $188.46 \pm 0.60$ | $\mathbf{49.75 \pm 0.18}$ |

Table 9: **Execution time of the proposed method**. Average run time and standard deviation in milliseconds on a $512 \times 512$ color image averaged over 20 runs for different operators. Timings are reported for an NVIDIA RTX 4090 GPU. Unlike DDRM and (u)DPIR, the proposed model does not require running multiple iterations, and is $3.7\times$ faster than the unrolled counterpart uDPIR, and $120\times$ faster than the DDRM diffusion method.

## F ADDITIONAL RESULTS ON MEDICAL IMAGING TASKS

**Single coil MRI** We provide in Figure 9 visual results for MRI reconstructions on acceleration factors $\times 4$ and $\times 8$ respectively.

**Multi-coil MRI** We provide in Figure 11 results on the multi-coil MRI inverse problem where we simulate $L = 15$ coil maps. The UNet reconstruction shows a less smooth aspect, penalizing PSNR.

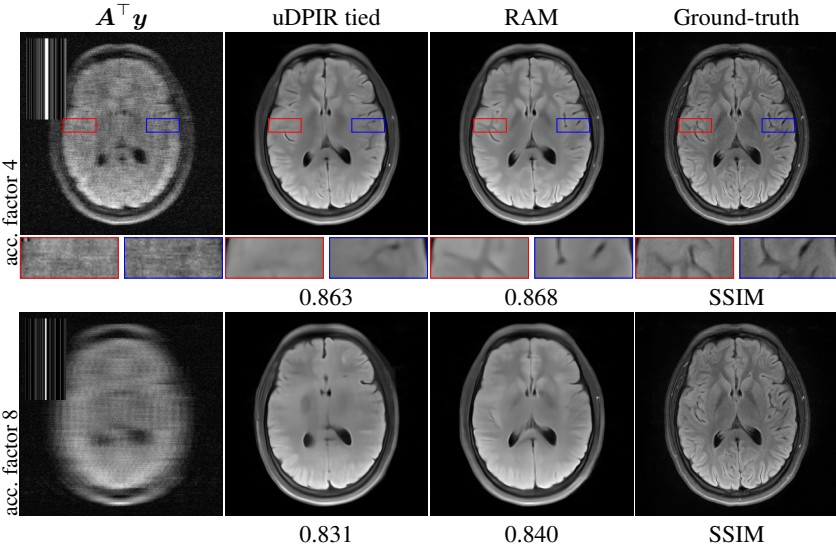

Figure 9: **Results on MRI for acceleration factors 4 and 8.** The Fourier mask is shown in the top left corner of the backprojection.

**CT** We provide in Figure 10 visual results for computed tomography for the in-distribution setup on the top row (i.e., with a setup similar to that of training), and the out-of-distribution setup on the bottom row. In the latter case, measurements are degraded with additional Poisson noise, unlike the training setup.

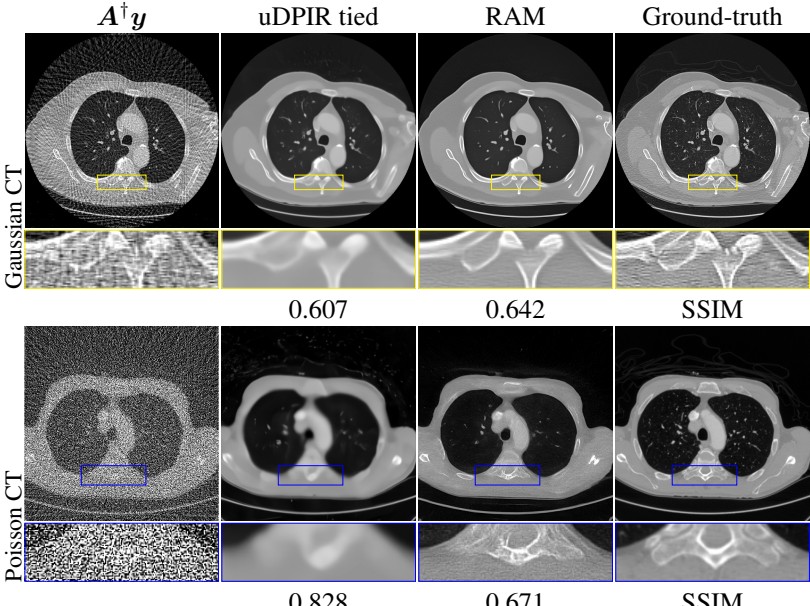

Figure 10: **Results on CT**. Top row: CT with Gaussian noise, similar to the training setup. Bottom row: CT with Poisson noise, unseen during training.

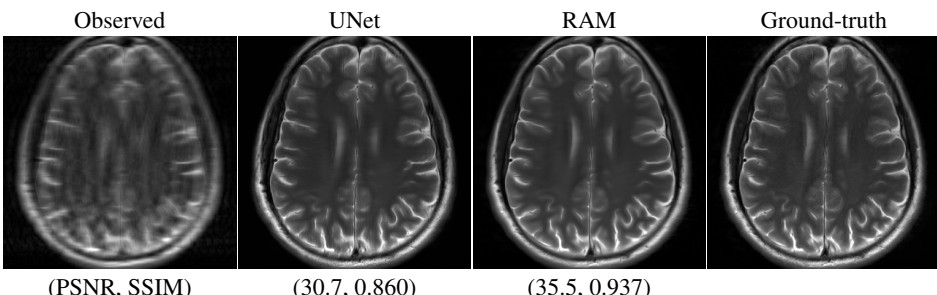

Figure 11: **Results on the multi-coil MRI problem** with acceleration factor 8. The UNet model is from (Zbontar et al., 2018).

## G  ADDITIONAL RESULTS ON NATURAL IMAGING TASKS

**Motion deblurring** We provide additional visual results for the motion-blur ("hard") task on DIV2K dataset samples in Figure 12. We observe that the proposed method performs on par with uDPIR with tied weights and DDRM, although some mild visual differences may be noticed. See, for instance, the windows in the blue zoombox of the church image, which appear to show a repetitive pattern in DDRM that is not faithful to the ground-truth image.

**Super-resolution** We provide visual results for bicubic super-resolution with factor 4 and with additive Gaussian noise of standard deviation $\sigma = 0.01$ on BSD68 dataset samples in Fig. 13. For this task, we also compare the proposed result with the state-of-the-art transformer-based SwinIR model (Liang et al., 2021) trained specifically for this degradation. We further compare with SR Neural Operator (SRNO) (Wei & Zhang, 2023). We observe that the proposed method outperforms the DPIR algorithm, but underperforms compared to SwinIR and SRNO.

**Grayscale Poisson-Gaussian denoising** We provide visual results for Poisson-gaussian denoising on a grayscale BSD68 dataset sample in Fig. 14. We follow the formulation from (7). On both Poisson denoising and Gaussian denoising, we observe that the model is relatively robust to out-of-distribution parameters. In both cases, while the reconstruction quality degrades as the degradation level increases, low-frequency features are preserved in the reconstruction, and no substantial artefacts are introduced.

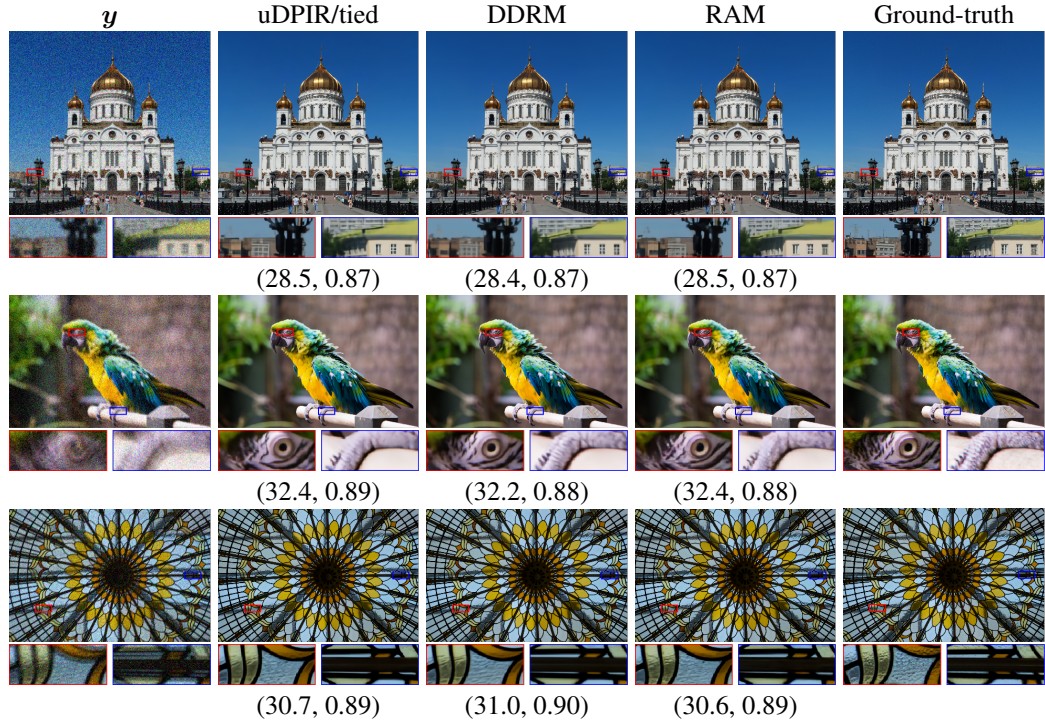

Figure 12: **Deblurring results** on the "Motion hard" problem, on samples from the DIV2K dataset. PSNR and SSIM metrics are provided below each reconstruction.

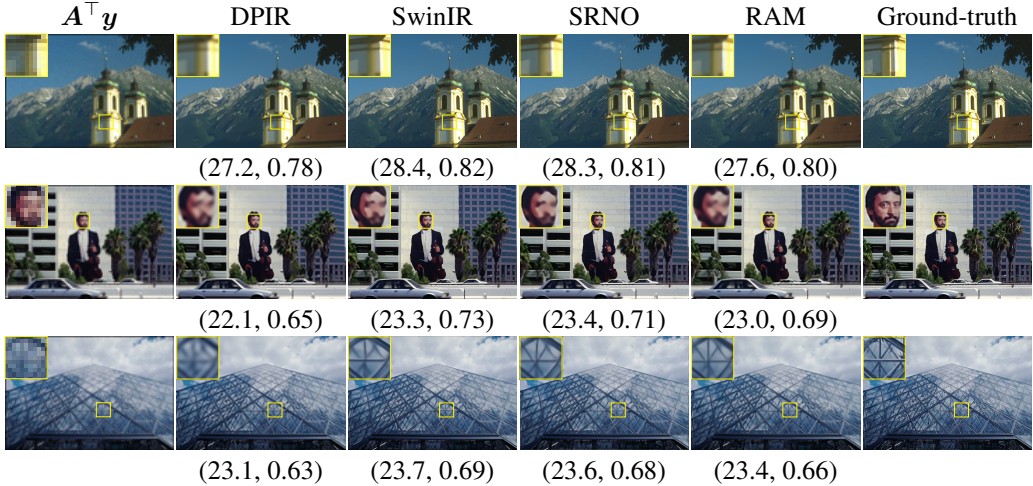

Figure 13: **Super-resolution results** on the SR×4 problem, on samples from the BSD68 dataset. (PSNR, SSIM) metrics are provided below each reconstruction.

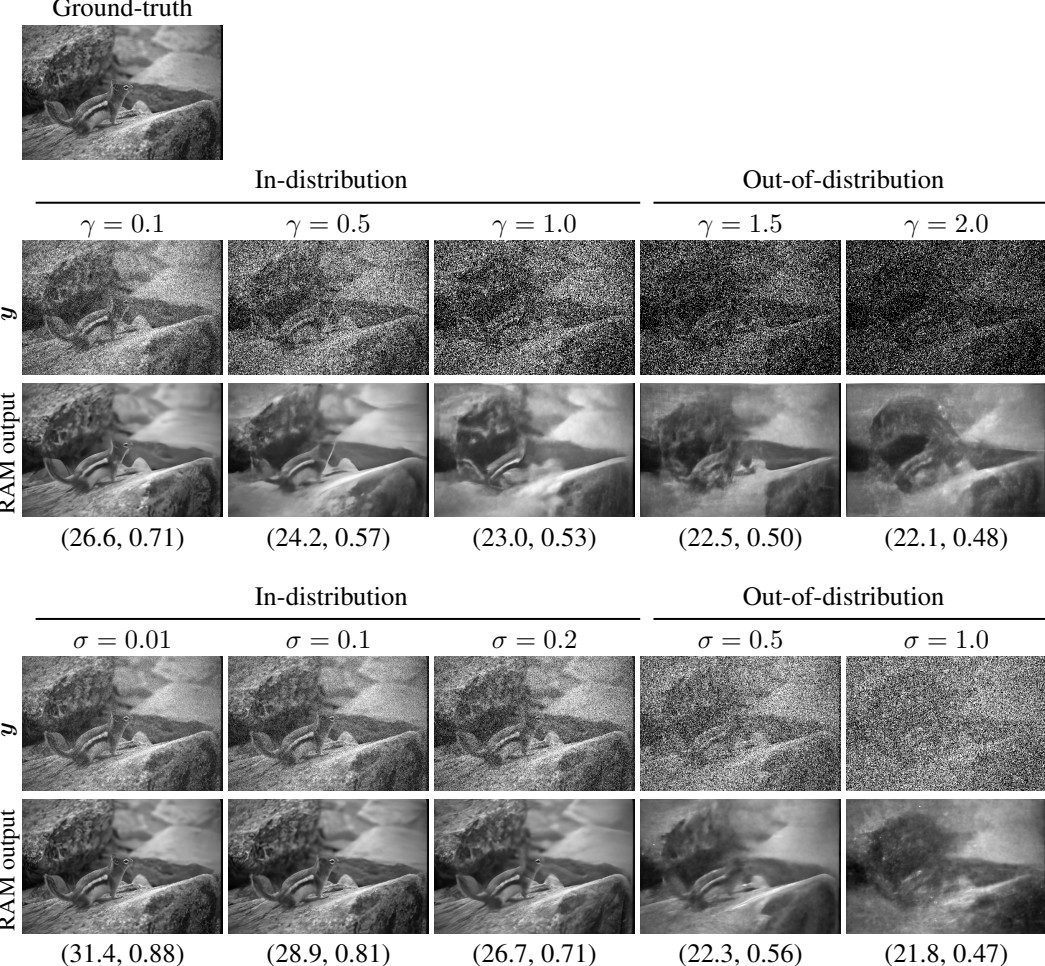

Figure 14: **Grayscale Poisson-Gaussian denoising** on a sample from the BSD68 dataset. (PSNR, SSIM) metrics are provided below each reconstruction. On the top row, we show input images and reconstruction for various $\gamma$ values in the Poisson-Gaussian model (7), with the Gaussian component fixed $\sigma = 0.01$. On the bottom row, we show input images and reconstruction for various $\sigma$ values in the Poisson-Gaussian model (7), with the Poisson component fixed $\gamma = 0.01$.

**Image inpainting** We provide visual results for inpainting with mild Poisson-Gaussian noise on a color BSD68 dataset sample in Fig. 15. We show results on inpainting masks applied either per-channel (top rows) or pixel-wise (bottom rows), for various masking levels. In both cases, Poisson-Gaussian noise is applied with $\sigma = 0.01$ and $\gamma = 0.01$. We observe that the model is more robust to out-of-distribution mask sparsity for channel-wise masks than for pixel-wise masks, where artifacts appear when fewer than 20% of pixels are observed.

**Blind deblurring** Figure 18 provides additional visual results of blind deblurring using the blur identification network from Carbajal et al. (2023), on a subset of 3 real motion blur images of the Kohler dataset (Köhler et al., 2012).

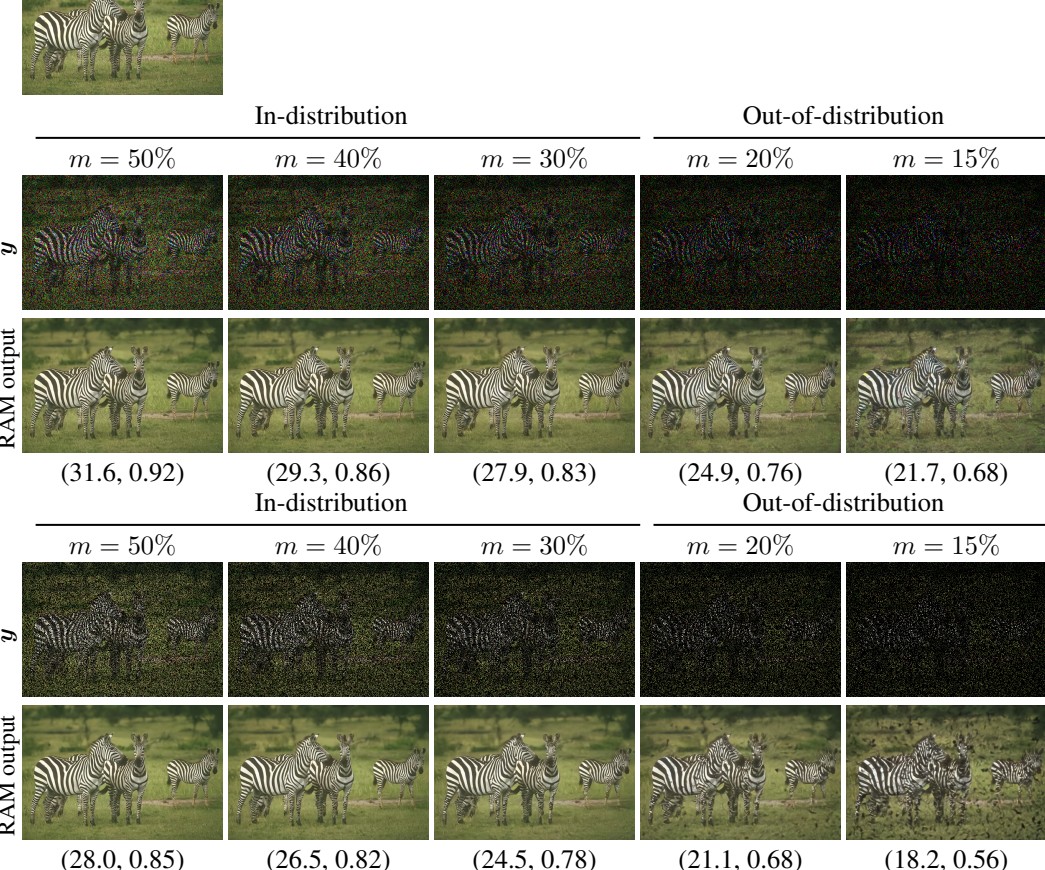

Figure 15: **Color inpainting with Poisson-Gaussian noise** on a sample from the BSD68 dataset. (PSNR, SSIM) metrics are provided below each reconstruction. Top rows: inpainting with per-channel masking, for various sparsity factors $m$ ($m = 30\%$ means that, for each channel, 30% of the pixels are kept). Bottom rows: inpainting with pixel-wise masking, for various sparsity factors $m$ ($m = 30\%$ means that 30% of the pixels are kept).

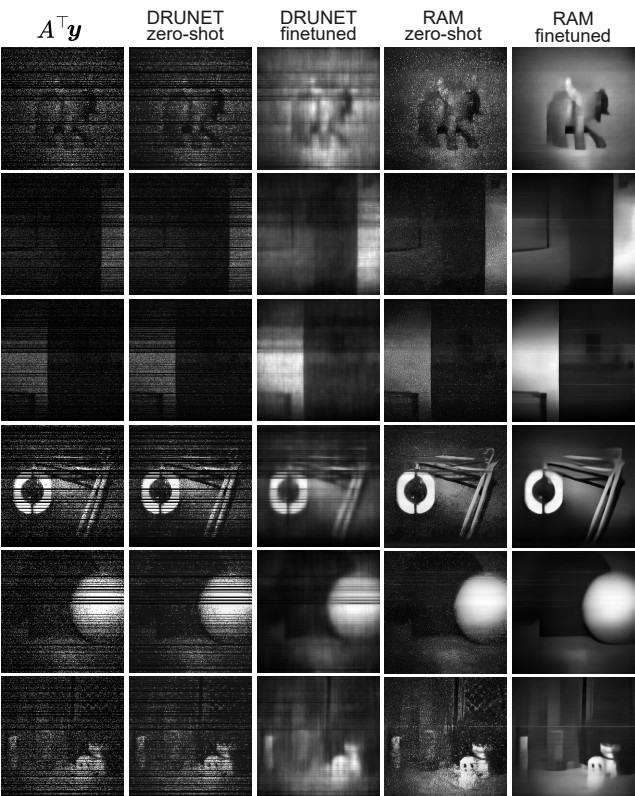

Figure 16: Additional LinoSPAD finetuning results.

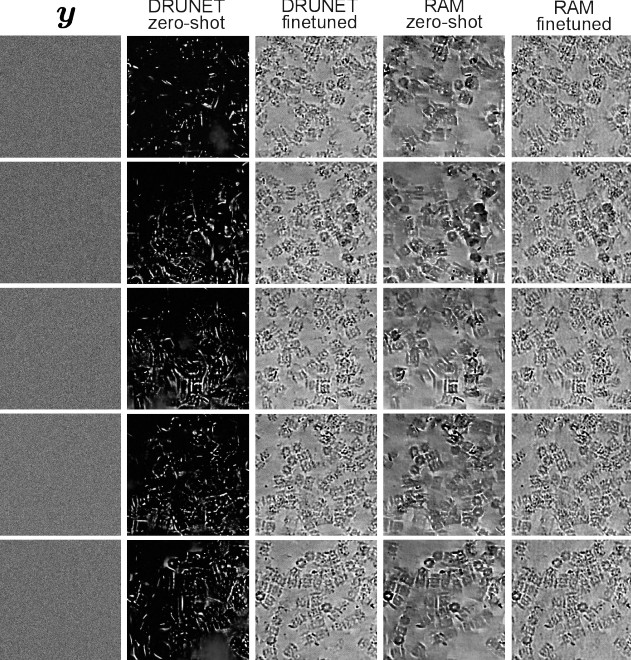

Figure 17: Additional Cryo-EM finetuning results.

|  $A^\dagger y$ | RAM | Ground-truth |

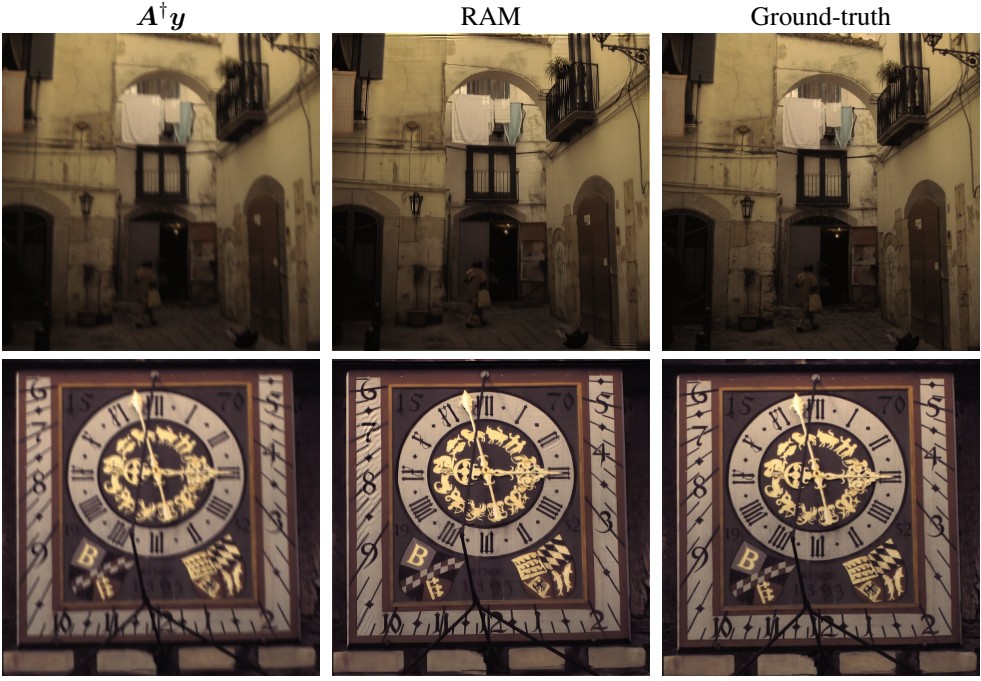

Figure 18: Additional blind deblurring results on the Kohler dataset.

## H  UNCERTAINTY QUANTIFICATION

We can obtain uncertainty estimates of the reconstructions obtained by RAM with the equivariant bootstrap algorithm (Tachella & Pereyra, 2024) described in Algorithm 1. This approach is similar to the standard parametric bootstrap (Efron, 1992), with the addition of a group of geometric transformations (set as random shifts, rotations, and flips in our experiments). The transforms are used to *probe* the confidence of the network on the reconstruction in the nullspace of the operator $A$.

We evaluate the uncertainty quantification algorithm on an inpainting task with $50\%$ missing pixels and Gaussian noise with standard deviation $\sigma = 0.02$, using the DIV2K validation dataset. As a baseline, we use the DDRM diffusion method, running the diffusion multiple times to obtain a set of approximate posterior samples. We obtain 100 samples for each test image, which requires 101 network evaluations for equivariant bootstrap and 1000 evaluations for DDRM (since each diffusion requires 100 denoiser evaluations). Estimated pixelwise errors, averaging over all color channels, are computed using $x_{\mathrm{err}}$ of Algorithm 1 and shown in Figures 7 and 20. We can see in both figures that the estimated errors follow closely the true errors.

We also evaluate the accuracy of these confidence regions by calculating the empirical coverage probabilities, as measured by the proportion of ground-truth test images that lie within the confidence regions for a range of specified confidence levels between 0% and 100%. This method provides a quantitative metric of the estimated uncertainty intervals (Thong et al., 2024). Results are shown in Figure 19. The proposed algorithm obtains good coverage without any additional calibration step, while the diffusion method provides highly overconfident intervals (note this behavior of diffusion models has been observed in various previous works (Tachella & Pereyra, 2024; Thong et al., 2024)).

---

**Algorithm 1** Equivariant Bootstrap

---

1: **Input:** Group $\mathcal{G} = \{g_1, \dots, g_{|\mathcal{G}|}\}$ acting on $\mathbb{R}^n$ via invertible maps $\boldsymbol{T}_g \in \mathbb{R}^{n \times n}$, RAM model R, number of bootstrap samples $N$.
2: Reconstruct $\hat{\boldsymbol{x}} = \mathrm{R}(\boldsymbol{y}, \boldsymbol{A}, \sigma, \gamma)$
3: **for** $i = 1, \dots, N$ **do**
4:     Draw $g_i \sim \mathrm{Unif}(\mathcal{G})$
5:     Generate bootstrap measurement

$$\tilde{\boldsymbol{y}}^{(i)} \sim \gamma \mathcal{P}(\frac{\boldsymbol{A}\boldsymbol{T}_{g_i}\hat{\boldsymbol{x}}}{\gamma}) + \sigma\boldsymbol{n}$$

with $\boldsymbol{n} \sim \mathcal{N}(\boldsymbol{0}, \boldsymbol{I})$.
6:     Compute the bootstrap replicate

$$\tilde{\boldsymbol{x}}^{(i)} = \boldsymbol{T}_{g_i}^{-1}\,\mathrm{R}(\tilde{\boldsymbol{y}}^{(i)}, \boldsymbol{A}, \sigma, \gamma)$$

7: **end for**
8: **Output:** Bootstrap sample $\{\tilde{\boldsymbol{x}}^{(1)}, \dots, \tilde{\boldsymbol{x}}^{(N)}\}$, and pixelwise errors $\boldsymbol{x}_{\mathrm{err}} = \frac{1}{N} \sum_{i=1}^{N} (\tilde{\boldsymbol{x}}^{(i)} - \hat{\boldsymbol{x}})^2$ where the squares are taken elementwise.

---

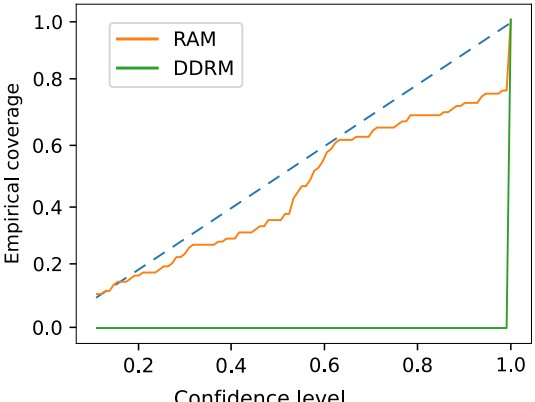

Figure 19: **Empirical coverage results.** Coverage of the proposed equivariant bootstrapping with RAM and of the posterior samples of DDRM (Kawar et al., 2022). A good coverage should follow the dotted line. The bootstrapping method provides significantly better uncertainty intervals than those computed using DDRM posterior samples.

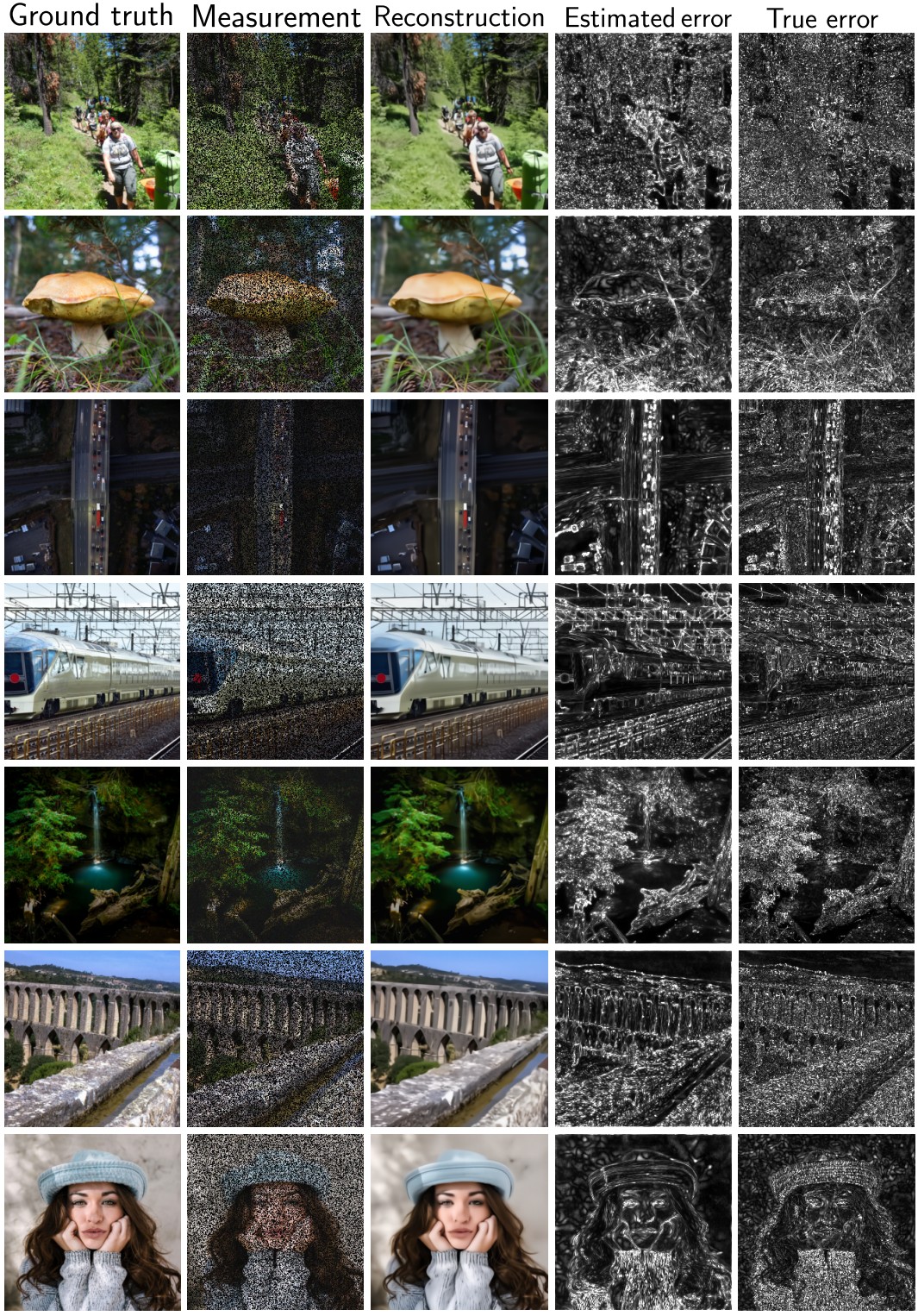

Figure 20: **Additional uncertainty quantification results.** Reconstructed images from DIV2K in-painting measurements with 50% of observed pixels and Gaussian noise of $\sigma = 0.02$. The estimated errors are computed using the equivariant bootstrap algorithm in Algorithm 1.

# I PHASE RETRIEVAL

We consider the phase retrieval problem $\boldsymbol{y} = |\boldsymbol{Bx}|$ where $\boldsymbol{x} \in \mathbb{C}$ is a complex image, and $\boldsymbol{B} = \boldsymbol{SF}\text{diag}(\boldsymbol{b})\boldsymbol{F}\text{diag}(\boldsymbol{a}) \in \mathbb{C}^{m \times n}$ is a computationally efficient approximation of an iid random matrix (see e.g. Oymak et al. (2018)) with $\boldsymbol{F} \in \mathbb{C}^{n \times n}$ being the fast Fourier transform, $\boldsymbol{a} \in \mathbb{C}^n$ and $\boldsymbol{b} \in \mathbb{C}^n$ being phase masks with random entries of unit absolute value and random phase (sampled uniformly between 0 and $2\pi$), and $\boldsymbol{S} \in \mathbb{C}^{m \times n}$ and subsampling or oversampling operator.

We evaluate Algorithm 2 on the set of 100 images in the DIV2K validation set, converting them to complex values by applying the preprocessing function $f(\boldsymbol{x}) = e^{\pi i \boldsymbol{x} / \|\boldsymbol{x}\|_\infty}$ where the exponential is computed elementwise. We use the cosine similarity as evaluation metric, since phase retrieval methods can only recover the signal up to a global phase (Dong et al., 2023). Figure 22 presents results for different sampling ratios $m/n$, showing that the proposed method can obtain robust results for $m/n < 1$, while standard methods only achieve good results for an oversampling ratio above $m/n \geq 4$ (Dong et al., 2023), and Figure 21 shows reconstruction results on a DIV2K sample in the case $m = n$. We use Algorithm 2 with $N = 40$ iterations, and noise parameters $\sigma = 0.3$ and $\gamma = 0.001$.

$\boldsymbol{A}^\dagger\boldsymbol{y}$        RAM        Ground-truth

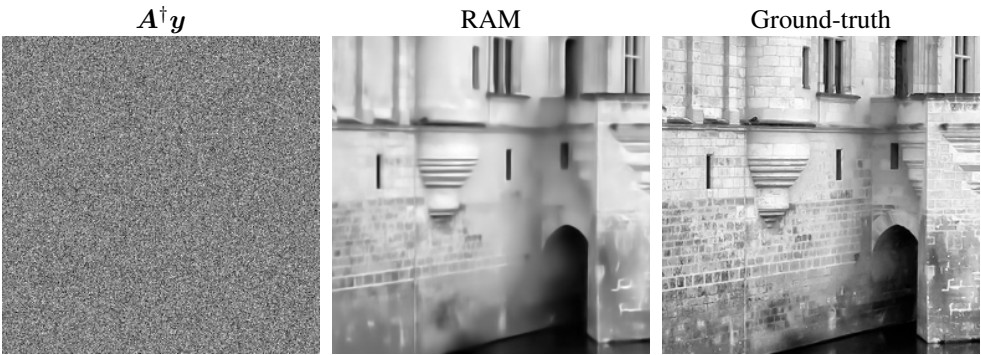

Figure 21: **Phase retrieval results** on a sample from the DIV2K validation set

---

**Algorithm 2** Phase retrieval with RAM

---

1: **Input:** Phaseless measurements $\boldsymbol{y} = |\boldsymbol{Bx}|$. Fixed $\gamma$ and $\sigma$.
2: Initialize the estimate at random $\hat{\boldsymbol{x}} \sim \mathcal{N}(\boldsymbol{0}, \boldsymbol{I})$.
3: **for** $i = 1, \ldots, N$ **do**
4:      Estimate measurement phase $\hat{\phi} = \text{phase}(\boldsymbol{B}\hat{\boldsymbol{x}})$
5:      Reconstruct $\hat{\boldsymbol{x}} = \text{R}(\boldsymbol{y}\exp(i\hat{\phi}), \boldsymbol{B}, \sigma, \gamma)$
6: **end for**
7: **Output:** Reconstruction $\hat{\boldsymbol{x}}$.

---

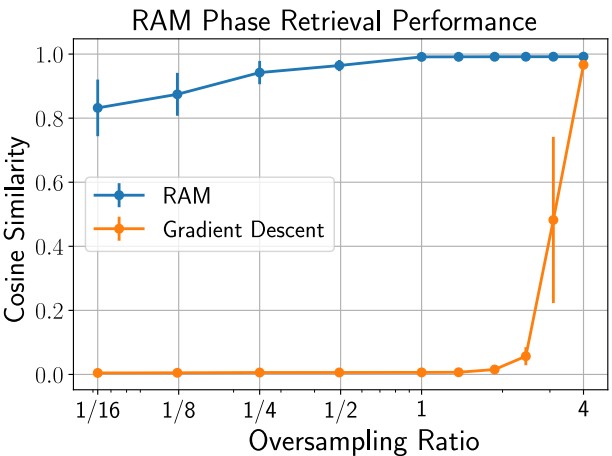

Figure 22: **Phase retrieval results for different sampling ratios.** Cosine similarity obtained by Algorithm 2 and gradient descent on the validation set of DIV2K images.

