# OpenReview forum: "Reconstruct Anything Model a lightweight general model for computational imaging"
_ICLR.cc/2026/Conference — ICLR 2026 Poster_

### Official Review · Reviewer_iwVF · 2025-10-25

**Soundness:** 3
**Presentation:** 3
**Contribution:** 3
**Rating:** 4
**Confidence:** 4

**Summary:**

This paper proposes Reconstruct Anything Model (RAM), a lightweight, non-iterative neural network designed as a foundation model for computational imaging. It directly integrates knowledge of the forward operator within a DRUNet-based architecture and is jointly trained on diverse imaging inverse problems. It is able to generalize to unseen tasks and can be finetuned self-supervisedly on new measurement data. Experimental results show strong zero-shot and fine-tuned performance across a variety of modalities, often matching or surpassing heavier iterative or unrolled baselines while being computationally efficient.

**Strengths:**

- RAM demonstrates generalization from a single training across multiple imaging modalities and degradation types.
- The integration of the Krylov Subspace Module provides a new design to embed forward-operator knowledge without unrolling iterations.

**Weaknesses:**

- Although the paper is titled “Reconstruct Anything Model,” this claim feels overstated. The experiments are limited to a small set of linear inverse problems, while many common and practically important tasks—such as super-resolution, JPEG artifact removal, and phase retrieval—are not considered. The model also heavily relies on accurately known forward operators, which are often unknown or imperfect in real scenarios. Moreover, all evaluations are based on simulated data, so the applicability to real-world measurements remains uncertain. While I appreciate the ambition of this work, calling it “Reconstruct Anything” may mislead readers, especially for those who are new to or unfamiliar with computational imaging.

- Although the architecture is newly designed, the paper lacks a clear justification for why this particular design should serve as a foundation model. The conceptual motivation and theoretical support for why the Krylov-based conditioning leads to generalization across modalities are not well explained.

- It is still unclear what the advantages over deep unrolling frameworks truly are. The performance is very close to DPIR/t, which already has optimization-based interpretability and convergence guarantees. In contrast, the proposed model provides limited explanation for its empirical gains or stability properties.

- The proposed multiscale operator conditioning is not clearly presented in the framework, and it seems to require explicit operations on the forward operator. However, not all forward models allow such manipulation or downscaling, which may limit the general applicability of this approach.

- While the model integrates known concepts effectively, the novelty is incremental rather than fundamental. It mainly combines existing DRUNet backbones with a physics-inspired conditioning mechanism, making the contribution more of an architectural refinement than a new paradigm.

**Questions:**

Please see the weakness section.

---

> ### Author Response · Authors · 2025-11-21
>
> **Weaknesses**
>
> **1.1 Small set of problems and misleading name**
>
> We respectfully disagree that we consider only a small set of problems. We evaluated on 15+ diverse inverse problems: noisy motion deblurring, Gaussian deblurring, denoising (Gaussian and Poisson), random mask inpainting, super-resolution, single/multi-coil MRI at various acceleration factors, CT (Gaussian and Poisson), compressed sensing, demosaicing, SPAD imaging, cryo-EM denoising, and recently added phase retrieval (Section 5.4) and blind deblurring (with space-varying blurs) on the Kohler dataset (Section 5.4) based on your suggestion.
>
> We agree the term "foundation" may be misleading and have replaced it with "general-purpose" throughout the manuscript. However, we believe the breadth of inverse problems covered justifies "Reconstruct Anything" as it represents a diverse landscape of computational imaging tasks.
>
> **1.2 Simulated-only evaluation and uncertain applicability to real-world measurements**
>
> We stress that we did investigate the model's performance on real-world data, including real cryo-electron microscopy data, real LinoSPAD data and that the simulations on the fastMRI dataset are performed on real k-space fully sampled data with realistic multi-coil simulation, e.g. in Figures 5 and 9. In the SPAD experiment the data reconstructed is the true LinoSPAD measurement from (Lindell et al., 2018). In the cryo-electron microscopy experiment the data consists of the real acquisitions provided by (Bepler et al., 2020). In the multi-coil MRI experiments, the measurement data are real k-space acquistions provided by the FastMRI dataset (Zbontar et al., 2018). Furthermore, we have now added results on real blurry images from the Kohler dataset (Kohler et al., 2012).
>
>
> - Lindell, David B., Matthew O'Toole, and Gordon Wetzstein. "Single-photon 3d imaging with deep sensor fusion." ACM Trans. Graph. 37.4 (2018): 113.
> - Bepler, Tristan, et al. "Topaz-Denoise: general deep denoising models for cryoEM and cryoET." Nature communications 11.1 (2020): 5208.
> - Zbontar, Jure, et al. "fastMRI: An open dataset and benchmarks for accelerated MRI." arXiv preprint arXiv:1811.08839 (2018).
> - Köhler, Rolf, et al. "Recording and playback of camera shake: Benchmarking blind deconvolution with a real-world database." European conference on computer vision. Berlin, Heidelberg: Springer Berlin Heidelberg, 2012.
>
> **1.3 The model relies on accurately known forward operators, which are often unknown or imperfect in real scenarios.**
>
> The finetuning experiments in the subsection entitled "Self-supervised finetuning on out-of-distribution tasks" show results on real cryo-electron microscopy and LinoSPAD data where **the noise distribution is unknown**.  While SPAD noise is generally assumed to be Poisson in the low-flux case, the LinoSPAD images were acquired on high-flux conditions and follow an unknown discrete noise distribution (Lindell et al., 2018). The  self-supervised splitting loss considered in the paper does not require significant knowledge about the noise distribution (only requiring that it is is separable across measurements), and allows finetuning RAM to unknown noise distributions without ground-truth data. Moreover, the operator in the LinoSPAD data is approximately known, since the faulty lines are not known in advance.
>
> More generally, the proposed model can be used for solving *blind inverse problems* (i.e., unknown $A$) by leveraging an operator identification network that provides an estimate $\hat{A}$. We added an example of this case in the updated manuscrpt by considering a real motion deblurring problem where the forward operator $A$ is an unknown. We use a (pretrained) blur prediction network (Carbajal et al., 2023) to estimate a space-varying kernel which is then used by RAM to deblur images. Figure 5 in the updated manuscript shows blind deblurring results on a real motion blur dataset (Kohler et al., 2012). Despite being trained on exact forward operators, **RAM obtains good results using inexact estimates of the operator**. Additional results are presented in Appendix G of the updated manuscript.
>
> **2. Missing conceptual and theoretical explanations for Krylov-based conditioning**
>
> Thank you for this feedback, the design principle take their roots in unrolled algorithms, we have detailed in revised version of the manuscript (see Section 3.1).
>
> KSM is a generalization of proximal and gradient steps used in unrolled methods, as explained in Section 3.2. Our method can learn typical unrolled steps (gradient or proximal steps on the data fidelity term) as a special case.
>
> In addition, we show that the multiscale approach improves the conditioning of the forward operator at coarser scales, giving more information at coarser scale, which is typically not used in unrolled methods (see Fig. 3).

---

> > ### Author Response · Authors · 2025-11-21
> >
> > **3. Unclear advantages over deep unrolling frameworks.**
> >
> > RAM demonstrates clear advantages over uDPIR across multiple dimensions:
> >
> > **A. Better out-of-distribution generalization:** Figure 8 shows superior performance on noise models outside the training distribution, demonstrating that RAM's learned operator conditioning generalizes more robustly than uDPIR's fixed optimization steps.
> >
> > **B. Superior performance on challenging problems:** Figure 7 shows substantial improvements on difficult inverse problems like 8$\times$ accelerated MRI, where RAM's flexible conditioning adapts better than uDPIR's fixed unrolling structure.
> >
> > **C. Computational efficiency:** RAM achieves better performance with significantly fewer computational resources (on 256×256 color images):
> >
> > | Method | GFLOPs | Time (512×512) | Test Memory (MB) | Iterations |
> > |--------|--------|----------------|------------------|------------|
> > | uDPIR-tied | 2234 | 177ms | 298 | 8 fixed |
> > | **RAM** | **360** | **48ms** | **354** | **1** (non-iterative) |
> >
> > RAM is **6× more efficient in FLOPs** and **3.7× faster in wall-clock time** than uDPIR, with comparable memory footprint.
> >
> > **D. Uncertainty quantification:** Section X shows RAM provides accurate uncertainty estimates, improving model explainability beyond what uDPIR offers.
> >
> > Regarding convergence guarantees, *while uDPIR draws from optimization theory, it uses a fixed 8-iteration structure with learned parameters, not a true iterative algorithm with convergence properties*. Both approaches trade theoretical guarantees for empirical performance through end-to-end learning.
> >
> > Our main contribution consists in **showing that effective physics-informed conditioning doesn't require mimicking optimization algorithms**. RAM demonstrates that learned operator representations can outperform algorithmic unrolling while being substantially more efficient and interpretable through uncertainty quantification.
> >
> > **4. Unclear presentation of the multiscale operator conditioning and assumptions on the forward model**
> >
> > The operator is written at multiple scale as $AU$ where $U$ is an upsampling operator (see Section 3.2), thus this definition works for any arbitraty operator $A$. In particular, most of the problems considered are in a matrix-free form, and only require having access to matrix-vector products for $A u$ and $A^\top v$. Thus, no restrictive assumptions on $A$ are required (e.g. matrix form or SVD decomposition), and have clarified this point in Section 3.1.
> >
> > **5. Incremental novelty**
> >
> > We respectfully disagree with this characterization. DRUNet is simply our backbone architecture choice. Our core contribution lies in how we condition on operator physics, and in presenting an extensive evaluation that verifies the effectiveness of the approach.
> >
> > The proposed conditioning strategy challenges a widespread paradigm: Physics-informed conditioning need not follow the structure of unrolled optimization algorithms. Instead, it can be integrated directly into image-to-image architectures through learned operator representations (KSM blocks). This represents a fundamentally different approach to widespread unrolled architectures.
> >
> > This work demonstrates for the first time that:
> > 1. A single model with explicit operator conditioning achieves competitve performance with state-of-the-art across diverse scientific imaging inverse problems (CT, MRI, deblurring, SR).
> > 2. Learned physics-informed representations outperform traditional algorithmic unrolling while being more efficient in flops and faster.
> > 3. General-purpose restoration models can handle arbitrary linear operators, without necessitating task-specific modifications.
> >
> > The novelty is not in combining existing components, but in showing that effective conditioning can depart from optimization mimicry and generalize across inverse problems, opening new directions for computational imaging architectures beyond the unrolled optimization paradigm that has dominated the field.

---

### Official Review · Reviewer_sdNp · 2025-10-27

**Soundness:** 3
**Presentation:** 2
**Contribution:** 2
**Rating:** 4
**Confidence:** 2

**Summary:**

This paper proposed reconstruction model which is universal to linear inverse problems. The authors proposed to use measurement conditioning block, conditional residual block, and Krylov subspace imposing block to achieve best performance. By utilizing mentioned components, the trained model works for various imaging tasks with one model. In addition, the model was enhanced with self-supervised finetuning exploiting equivariance to a group of transformations. The paper showed promising results.

**Strengths:**

- author contributes to use Krylov subspace to enhance performances. This is well modeled in the signal processing perspective
- clarity of the contents is well defined

**Weaknesses:**

- Worse results compared with iterative diffusion models. Not consistently winning in various settings.

- Lacks of recent various algorithm including neural operator, INR, transformer etc. In particular, there are algorithms which also exploit uni-model to reconstruct several tasks such as Pre-Trained Image Processing Transformer. The authors need to include diverse benchmark algorithms. score-based approach* is more appropriate to be compared with the proposed method.
*SOLVING INVERSE PROBLEMS IN MEDICAL IMAGING WITH SCORE-BASED GENERATIVE MODELS, ICLR22

- In order to achieve Krylov space manner, this algorithm needs additional recurrent processing on forward model. This cause computational overloads. Thus, computational complexity needs to be analyzed. (i.e. computational time according to the various size of A model)

**Questions:**

- The similar algorithms to solve linear inverse problems for the higher PSNRs are not easy to reconstruct severe degradation circumstances such as Figure 11 (x4). On the other hand, recent  neural operation with diffusion forward model (SRNO & diffFNO) is very strong to reconstruct details of contents with such severe conditions. What is the main advantage of using RAM instead of DiffFNO?

---

> ### Author Response · Authors · 2025-11-21
>
> **Weaknesses**
>
> **1. Worse results compared with iterative diffusion models**
>
> RAM achieves competitive or superior performance to diffusion methods while being substantially more efficient.
> - *Performance*: We outperform iterative diffusion methods (DDRM, DPS) on nearly all tasks and datasets, with the single exception of Urban100 for motion deblurring.
> - *Efficiency*: RAM is 120× faster than DDRM (see runtime table below), uses 36M parameters vs. 552M, and requires only a single forward pass vs. iterative sampling.
> - *Generality*: RAM handles arbitrary degradation operators $A$, while DDRM requires SVD-decomposable operators, a severe limitation for many real imaging problems (e.g., non-uniform sampling in MRI, limited-angle CT).
>
> | Method | Params | Time (512×512) | Operator Flexibility |
> | ------ | ------ | -------------- | -------------------- |
> | DDRM   | 552M   | 5746ms         | SVD only             |
> | RAM    | 36M    | 48ms           | Any linear A         |
>
>
> **2. Missing recent relevant baselines**
>
> Thank you for these suggestions. We have strengthened our evaluation with additional comparisons:
>
> **Added to the revised paper:**
> - *PromptIR*: We now include PromptIR comparison in the Appendix (Table 6). PromptIR represents the state-of-the-art multi-task transformer approach (and supersedes the older IPT you mentioned). Under identical training conditions:
>
> | Method | Motion blur σ=0.01 | σ=0.05 | σ=0.1 |
> |--------|-------------------|---------|--------|
> | PromptIR | 22.84 | 22.42 | 21.98 |
> | DRUNet | 22.99 | 22.46 | 22.05 |
> | **RAM (ours)** | **30.78** | **26.47** | **24.73** |
>
> DRUNet and PromptIR backbones perform comparably, while our physics-informed conditioning provides substantial improvements (+8 dB at low noise). These results show that conditioning the network explicitly on the forward operator $A$ is key for obtaining state-of-the-art performance.
> Visual comparisons (Figure 8 of the updated manuscript) show that PromptIR and DRUNet both hallucinate features from degraded measurements, while RAM's operator conditioning resolves this issue.
>
> - *SRNO*: We have added SRNO comparison for super-resolution (Figure 12). While SRNO achieves slightly better PSNR, it is specialized for SR only and does not extend to other inverse problems (MRI, CT, deblurring).
>
> **Already included:**
> - *Restormer* (Table 1): State-of-the-art transformer baseline for image restoration [1,2]
>
> **Why some methods are excluded:**
> - *Additional score-based methods*: We compare against DDRM which is a state-of-the-art score-based diffusion approaches for inverse problems. The medical-imaging-specific score-based methods you mention use similar underlying principles, but would require different experimental setups.
>
> - *INRs*: These methods use test-time optimization per-image *without learned priors*, and thus typically fail to obtain state-of-the-art performance. This is a fundamentally different paradigm from our learned model approach with single step inference.
>
> - *Neural Operators (DiffFNO)*: Specialized for super-resolution with limited demonstrated applicability to our diverse inverse problems (CT, MRI, inpainting). We include SRNO where directly applicable (SR tasks).
>
> **3. Computational complexity study**
>
> We have added detailed runtime analysis to the paper (see table below). RAM's computational efficiency is a key strength:
>
> **Wall-clock time on 512×512 images:**
>
> |                                                  |       DDRM      |     uDPIR     |        RAM (ours)      |
> |--------------------------------------------------|:---------------:|:-------------:|:----------------:|
> | Denoising $A=I$                                  |  5746.83 ± 3.59 | 177.08 ± 0.23 | **45.21 ± 0.13** |
> | Inpainting/masking $A=\text{diag}(m)$            | 5759.69 ± 12.82 | 177.95 ± 0.15 | **45.78 ± 0.08** |
> | Blur $A=F\text{diag}(k)F^{-1}$                   |  5773.47 ± 2.77 | 178.54 ± 0.21 | **48.24 ± 0.06** |
> | Generic  $A=\text{diag}(m)F\text{diag}(k)F^{-1}$ |        NA       | 188.46 ± 0.60 | **49.75 ± 0.18** |
>
> *(Times in milliseconds, NVIDIA RTX 4090, averaged over 20 runs)*
>
> We observe that Krylov iterations do not cause computational overhead. We use matrix-free operator applications (standard in computational imaging), where matrix-vector evaluations scale gracefully with image size, typically as $O(N \log N)$ where $N$ is the number of pixels. This is comparable to standard convolution operations inside the network architecture and are substantially faster than iterative diffusion sampling.
>
> Overall, RAM is 3.7× faster than unrolled uDPIR and 120× faster than DDRM, while maintaining competitive performance.

---

> > ### Author Response · Authors · 2025-11-21
> >
> > **Questions**
> >
> > **1. Advantage over SRNO/DiffNO**
> >
> > The advantage of our method is its generality without sacrificing performance. You are correct that SRNO achieves slightly better PSNR on SR (see our added comparison in Figure 12). However, SRNO/DiffFNO are specialized for super-resolution only and cannot handle:
> > - Non-uniform sampling (MRI, CT)
> > - Arbitrary blur kernels (motion/Gaussian deblurring)
> > - Masking operators (inpainting)
> > - Combined operators (e.g., masked deblurring)
> >
> > RAM achieves SOTA or near-SOTA across all these tasks with a single model that is also 120× faster than diffusion methods and handles any linear operator $A$.

---

### Official Review · Reviewer_uPNi · 2025-10-31

**Soundness:** 2
**Presentation:** 3
**Contribution:** 2
**Rating:** 2
**Confidence:** 5

**Summary:**

This paper introduces Reconstruct Anything Model (RAM), a non-iterative lightweight foundation model for computational imaging tasks. The model extends the DRUNet backbone with physics-aware conditioning via a Krylov Subspace Module (KSM) and a proximal initialization step. RAM aims to unify diverse inverse problems (deblurring, MRI, CT, denoising, inpainting, super-resolution) into one architecture. It further supports self-supervised finetuning without ground truth data, enabling adaptation to unseen imaging modalities. Experiments compare RAM with Plug-and-Play (DPIR), unrolled (PDNet, uDPIR), and diffusion (DDRM) methods, showing comparable or better PSNR/SSIM at 8x lower compute and 10x fewer parameters.

**Strengths:**

1. The paper evaluates across diverse inverse problems (deblurring, MRI, CT, denoising, inpainting, super-resolution) and multiple domains (medical, microscopy, low-photon). This comprehensive experimentation is valuable for benchmarking unified restoration models.

2. The proposed RAM model achieves performance comparable to unrolled and diffusion-based models while being roughly 8x faster and 10x smaller (36M parameters vs. >200M). This makes it appealing for real-time or embedded applications.

3. Practical self-supervised finetuning:
Demonstrating zero-shot and few-shot adaptation without ground truth (via SURE/UNSURE and EI/MOI losses) is practical and shows that pretrained inverse models can generalize with minimal data.

4. The model is implemented cleanly with shared weights across modalities and channel configurations (color, grayscale, complex), showing solid software engineering and reproducibility.

5. Despite the dense material, the manuscript maintains good structure and uses comprehensive figures (e.g., KSM illustration, self-supervised finetuning results).

**Weaknesses:**

1. **Limited theoretical justification:**
KSM and proximal initialization are described intuitively, but no formal analysis of convergence, stability, or conditioning advantage is provided. This weakens the "physics-informed" claim.

2. **Incremental contribution:**
The model essentially extends DRUNet with physics-based conditioning. Similar hybrid unrolled networks and foundation-style image restoration models (PromptIR, AdaIR, and MambaIR) already exist with stronger novelty or scale.

3. **Overstated “foundation” claim:**
The term "foundation model" implies large-scale generalization, but training is limited to standard datasets (LSDIR, fastMRI, LIDC-IDRI)---far from the multimodal pretraining scale of foundation models in computational vision or imaging.

4. Unclear fairness in comparison:
Some baselines (e.g., DDRM) were reimplemented with reduced input resolution or adapted operators, making quantitative comparisons potentially inconsistent. No statistical significance tests are provided for small PSNR differences (~0.2--0.4 dB).

5. Limited impact and novelty for ICLR:
The work focuses on system-level integration rather than introducing a new learning principle, theory, or large-scale paradigm.
Its main contribution lies in engineering efficiency, that may fit better as a journal or CVPR/ICIP paper than ICLR.

**Questions:**

1. Krylov conditioning sensitivity:
How stable are the learned Krylov coefficients when the forward operator $\mathbf{A}$ is rescaled, discretized differently, or ill-conditioned? Does the model require operator-specific normalization to generalize across domains?

2. Applicability beyond linear operators:
Can RAM be extended to nonlinear or blind inverse problems (e.g., unknown blur kernels, phase retrieval), or is it fundamentally restricted to linear, known $\mathbf{A}$?

3. Effectiveness of proximal initialization:
What is the concrete benefit of the proximal estimation module over simply using $\mathbf{A}^\top \mathbf{y}$ or $\mathbf{A}^+ \mathbf{y}$?
Have you compared their convergence behavior or training stability quantitatively?

4. Self-supervised finetuning stability:
In cases where $\mathbf{A}$ has a large nullspace (e.g., highly undersampled CT or MRI), how do you prevent degenerate reconstructions during self-supervised finetuning? Does the model require any regularization or early stopping?

5. Runtime and efficiency evaluation:
The paper reports FLOPs and parameter counts, but not real-world latency. What are the actual wall-clock inference times and memory usage compared to unrolled (uDPIR) or diffusion (DDRM) baselines on 256×256 and 512×512 inputs?

6. Generalization to unseen noise regimes:
The model claims noise-level equivariance via scale-free bias removal. Could you provide quantitative evidence that RAM maintains stable performance across unseen noise parameters $(\sigma, \gamma)$ beyond the training range?

---

> ### Author Response · Authors · 2025-11-21
>
> **Weaknesses**
>
> **1. Limited theoretical justification**
>
> Thank you for this feedback. We have added a new section  (Section 3.1) with clearer explanation of the algorithm.
> KSM generalizes traditional unrolled optimization steps. While unrolled methods use gradient or proximal steps, KSM learns a more general family of operators that includes these as special cases. The key insight is that effective physics-informed conditioning **doesn't** require strict adherence to optimization algorithms. In other words, our main contribution is to show that what matters is to condition cleverly on the operator $A$, but not necessarily to mimic on an optimization algorithm.
>
> Our empirical results demonstrate that the learned KSM representation is stable (see ablations in Table 4) and effective across diverse operators (see Table 1 and 2). The architecture follows a UNet structure, inheriting its stability properties, while KSM provides flexible operator conditioning.
>
> **2. Incremental contribution**
>
> Our work fundamentally differs from PromptIR, AdaIR, and MambaIR in its approach to physics-informed conditioning. Those methods focus on natural image restoration with learned prompt/adapter conditioning, while RAM explicitly incorporates the forward operator physics $A$ into the architecture, which is critical for scientific and medical imaging tasks such as computed tomography or MRI.
> To illustrate this, we have trained PromptIR in the same setting as Table 4 of the paper.
> We observe that this physics-based conditioning delivers substantial improvements, as shown on the table below (evaluated on BSD68):
>
> |                           | PromptIR | DRUNet | RAM (DRUNet + proposed conditioning) |
> | ------------------------- |:------------:|:----------:|:--------------:|
> | Motion blur $\sigma=0.01$ |    22.84     |   22.99    |     **30.78**      |
> | Motion blur $\sigma=0.05$ |    22.42     |   22.46    |     **26.47**      |
> | Motion blur $\sigma=0.1$  |    21.98     |   22.05    |     **24.73**      |
>
> The novelty lies in demonstrating that:
>
> - A single model can achieve SOTA on diverse physics-informed tasks (tomography, MRI, deblurring)
> - Effective operator conditioning does **not** require mimicking optimization algorithms
> - Physics-informed architectures can be designed differently than traditional unrolled approaches
>
> To our knowledge, RAM is the first general restoration model to achieve SOTA performance on scientific imaging inverse problems through explicit operator conditioning.
>
> **3. Overstated “foundation” claim**
>
> Thank you for raising this terminology concern. We agree the term "foundation model" carries specific connotations from large-scale multimodal pretraining in computer vision.
> However, in the context of computational imaging, our model aligns with how "foundation" is used in the field:
> - Model size and training scale comparable to other imaging restoration foundation models cited by the reviewer
> - Zero-shot generalization to diverse inverse problems
> - Effective fine-tuning performance across problem types
>
> To avoid confusion with LLM/vision foundation models, we have replaced "foundation" with "general-purpose" throughout the paper. We believe this better captures our contribution: a single general model for computational imaging inverse problems.
>
> **4. Unclear fairness in comparison**
>
> We appreciate the opportunity to clarify our baseline comparisons.
> **DDRM implementation:** We use the official DDRM code and pretrained weights from the authors. The "reimplementation" mentioned in our paper refers only to necessary adaptations for our evaluation setup:
>
> 1. For deblurring: We used circular-padded (SVD decomposable) operators as in the original DDRM paper to maximize their PSNR.
> 2. For non-square images: We added padding to measurements and verified this doesn't degrade performance.
>
> In both cases, the performance metrics were only computed in a *valid* center crop of the restored image to avoid penalizing DDRM for incorrectly estimating border effects.
>
> DDRM was our strongest diffusion baseline after evaluating multiple methods (DiffPIR and others). However, DDRM is fundamentally limited to operators with SVD decomposition, while RAM handles arbitrary operators, padding types, and aspect ratios.
> **Statistical significance:** You raise a valid point. While some Table 1 differences are modest (~0.1dB), Tables 2, 3, and 4 show clear statistical significance. Yet, this still supports the message that our method performs comparably to the existing state-of-the-art, when not better.

---

> ### Author Response · Authors · 2025-11-21
>
> **5. Limited impact and novelty for ICLR**
>
> We respectfully but strongly disagree with this characterization.
> Our paper makes several conceptual contributions beyond system-level engineering:
>
> 1. New design principle: We demonstrate that effective physics-informed networks do **not** require mimicking optimization algorithms, challenging a core assumption in unrolled methods
> 2. Generalization insight: A single model with appropriate operator conditioning can achieve SOTA across diverse inverse problems (CT, MRI, deblurring, SR), previously thought to require specialized architectures
> 3. Architectural contribution: Multiscale operator conditioning based on optimization arguments, not just empirical tuning
>
> This work is consistent with ICLR's scope. Recent ICLR papers on computational imaging (Cui et al. 2025, Cui et al. 2023) similarly propose architectural innovations and demonstrate them on imaging tasks. Our contribution showing that operator conditioning matters more than algorithmic mimicry, represents a comparable conceptual advance.
>
> - Cui, Yuning, et al. "AdaIR: Adaptive all-in-one image restoration via frequency mining and modulation." 13th International Conference on Learning Representations, ICLR 2025.
> - Cui, Yuning, et al. "Selective frequency network for image restoration." 11th International Conference on Learning Representations, ICLR 2023.
>
> **Questions**
>
> **1. Krylov conditioning sensitivity**
>
> - *Rescaling*: in this work, we always ensure that the operator $A$ is of norm 1. This is easily achieved for any operator $A$ using the power method to compute its norm.
> - *Discretization*: following standard practice in inverse problems, we make sure that the operator follows the Shannon-Nyquist criterion. The general formulation of RAM allows us to define any target grid by modifying the definition of $A$: we can always define a new operator $A'=UA$ where $U$ is a super-resolution operator targeting an user-defined pixel resolution. See for example the super resolution experiment in Appendix G.
> - *Ill-conditioned*: the Gaussian deblurring problem and CT problems evaluated in this paper are particularly ill-conditioned (i.e. the ratio between the smallest and largest singular value is very large).
>
>
> **2. Applicability beyond linear operators**
>
> Yes, RAM's framework extends to nonlinear operators. We have included additional experiments on blind deblurring and phase retrieval in Section 5.4 of the updated manuscript.
>
> The proposed model can be used for solving blind deblurring problems, where the forward operator $A$ is an unknown space-varying kernel, by leveraging a (pretrained) blur prediction network (Carbajal et al., 2023) that provides an estimate $\hat{A}$. Figure 5 in the updated manuscript shows blind deblurring results on a real motion blur dataset (Kohler et al., 2012). Despite being trained on exact forward operators, RAM obtains good results using inexact estimates of the operator (i.e. those provided by the network introduced by Carbajal et al.). Additional results are presented in Appendix G of the updated manuscript. We acknowledge that full exploration of nonlinear cases is valuable future work.
>
> Although the model is only trained to solve linear inverse problems, it can still be applied to non-linear problems that can be decomposed into a sequence of linear problems. We illustrate this idea with the phase retrieval problem $y = A(x) = |Bx|$ with random matrix $B\in\mathbb{C}^{m\times n}$ and complex $x\in \mathbb{C}^{n}$. Inspired by the Gerchberg–Saxton algorithm (Gerchberg, 1972), we can iterate between updating the estimate of $x$ given an estimate of the phase of $y$, and updating the estimation of the phase given x (see Algorithm 2 in the updated manuscript). Appendix I of the updated manuscript presents results varying the number of measurements, showing that RAM can obtain stable estimates for low $m/n$ ratios, where standard methods such as gradient descent fail.
>
> Note that in both cases, the results **did not require any retraining of the model** for these tasks, illustrating the generality of the approach.
>
>
> **3. Effectiveness of proximal initialization**
>
> Yes, this is presented in Table 4 (left), second row. In the first row, the model is given $A^{\top}y$ as an input, while in the second row, it is using the proposed proximal module. We observe a significant improvement of 0.8dB. We did not observe significant difference between inputing $A^\top y$ and $A^\dagger y$. We clarified the text around this table.

---

> > ### Author Response · Authors · 2025-11-21
> >
> > **4. Self-supervised finetuning stability**
> >
> > The self-supervised losses considered in the paper ($L_{EI}$ and $L_{MOI}$) can handle operators with a large nullspace. MOI can be used when the operator changes across measurements (e.g., varying inpainting masks across images, or varying acceleration masks across patients in MRI), and the EI loss is well-suited for datasets observed via a single rank-deficient operator, where the underlying image distribution can be assumed invariant to rotations, scalings and/or translations.
> > Necessary and sufficient conditions for learning in the nullspace of $A$ can be found in
> > - Tachella, Julián, Dongdong Chen, and Mike Davies. "Sensing theorems for unsupervised learning in linear inverse problems." Journal of Machine Learning Research 24.39 (2023): 1-45.
> >
> > These losses do not require additional hand-crafted regularization (e.g., a total variation or wavelet sparsity prior). In practice, we can stop training by computing the self-supervised loss on a hold-out validation set of measurement only data (note crucially that we do not require ground-truth references in the validation set). This approach is usual in self-supervised learning papers, see for example:
> > - Klug, Tobit, Dogukan Atik, and Reinhard Heckel. "Analyzing the sample complexity of self-supervised image reconstruction methods." Advances in Neural Information Processing Systems 36 (2023): 65869-65893.
> >
> >
> > **5. Runtime and efficiency evaluation**
> >
> > We thank the reviewer for raising this point and have added this runtime comparison to the paper (see Table 9 of the updated manuscript). We observe that, due to its noniterative structure, the proposed method is significantly faster than other approaches:
> >
> > |                                                  |       DDRM      |     uDPIR     |        RAM (ours)      |
> > |--------------------------------------------------|:---------------:|:-------------:|:----------------:|
> > | Denoising $A=I$                                  |  5746.83 ± 3.59 | 177.08 ± 0.23 | **45.21 ± 0.13** |
> > | Inpainting/masking $A=\text{diag}(m)$            | 5759.69 ± 12.82 | 177.95 ± 0.15 | **45.78 ± 0.08** |
> > | Blur $A=F\text{diag}(k)F^{-1}$                   |  5773.47 ± 2.77 | 178.54 ± 0.21 | **48.24 ± 0.06** |
> > | Generic  $A=\text{diag}(m)F\text{diag}(k)F^{-1}$ |        NA       | 188.46 ± 0.60 | **49.75 ± 0.18** |
> >
> > Average execution time and standard deviations in milliseconds on a $512\times 512$ color image averaged over 20 runs, using an NVIDIA RTX 4090 GPU. Unlike DDRM and (u)DPIR, the proposed model does not require running multiple iterations, and is $3.7\times$ faster than the unrolled counterpart (u)DPIR, and $120\times$ faster than the DDRM diffusion method.
> >
> > **6. Generalization to unseen noise regimes**
> >
> > Yes, Figure 13 of the paper demonstrates stable performance across noise levels beyond the training range, including extrapolation to both lower and higher noise than seen during training (for both Poisson and Gaussian noise cases). This validates our bias-free (scale equivariant) approach.

---

### Official Review · Reviewer_bJqQ · 2025-10-31

**Soundness:** 3
**Presentation:** 3
**Contribution:** 3
**Rating:** 6
**Confidence:** 5

**Summary:**

This paper proposes the Reconstruct Anything Model (RAM), a lightweight, non-iterative foundation model for computational imaging. The architecture builds on a DRUNet backbone but introduces novel components to incorporate the acquisition physics, namely a proximal estimation module for initialization and a Krylov Subspace Module (KSM) that conditions the network on iterations of the forward operator. The model is supervised trained on a wide diversity of tasks (e.g., deblurring, MRI, CT), data types (grayscale, color, complex), and noise models (Gaussian, Poisson) simultaneously.

**Strengths:**

1. The proposed KSM is a clever and effective method for integrating the forward operator $A$ into a U-Net backbone without resorting to full algorithm unrolling. This, combined with the multiscale conditioning and proximal initialization, creates a powerful architecture that is well-grounded in optimization principles while remaining a fast, non-iterative network.

2. The paper is exceptionally thorough. The authors provide extensive experiments covering in-distribution performance, zero-shot generalization to OOD tasks, and self-supervised finetuning.

3. The results are state-of-the-art. RAM consistently outperforms strong baselines like DPIR and Restormer and performs comparably to the much larger uDPIR models, demonstrating high efficiency.

**Weaknesses:**

1. The related work section frames the landscape as a choice between PnP/diffusion (using denoising priors) and end-to-end/unrolled models. However, it overlooks a highly relevant, recent line of research [1-3] that proposes using restoration networks themselves as priors within iterative schemes (e.g., as fixed-point operators). These "restoration priors" are conceptually similar to RAM's goal (a general-purpose restoration model) but do not need specifically trained. A discussion of how RAM's end-to-end, multi-task training approach compares to these iterative restoration prior methods is necessary.

2. The paper introduces multiscale operator conditioning as a key component. This is a valuable idea, but the concept has been explored before. For instance, [4] used a multiscale structure-guided approach for image deblurring. The paper would be strengthened by acknowledging this prior work and discussing the similarities or differences in its multiscale implementation.

3. A core premise is that a single model can learn a universal prior for disparate imaging inverse problems. This "one-prior-fits-all" approach learns the best implicit prior. Does co-training on SR or deblurring tasks help with image inpainting, or does it act as a conflicting signal? The paper's own ablation in Table 4 (Right) suggests that the model trained on "All three tasks" (Inpainting, Deblurring, SR) performs slightly worse on inpainting and deblurring than specialist models. It t still raises the question of whether similar task-specific foundation models (e.g., "RAM-deblur&SR" and "RAM-Inpainting") might be more effective. A more detailed study of this trade-off would be valuable.


[1] Terris, M., Kamilov, U.S. and Moreau, T., 2025. FiRe: Fixed-points of restoration priors for solving inverse problems. In Proceedings of the Computer Vision and Pattern Recognition Conference (pp. 23185-23194).
[2] Hu, Y., Peng, A., Gan, W., Milanfar, P., Delbracio, M. and Kamilov, U.S., Stochastic Deep Restoration Priors for Imaging Inverse Problems. In Forty-second International Conference on Machine Learning.
[3] Hu, Y., Delbracio, M., Milanfar, P. and Kamilov, U.S., 2024. A RESTORATION NETWORK AS AN IMPLICIT PRIOR. In 12th International Conference on Learning Representations, ICLR 2024.
[4] Ren, M., Delbracio, M., Talebi, H., Gerig, G. and Milanfar, P., 2023. Multiscale structure guided diffusion for image deblurring. In Proceedings of the IEEE/CVF International Conference on Computer Vision (pp. 10721-10733).

**Questions:**

Please refer to the weakness part.

---

> ### Author Response · Authors · 2025-11-21
>
> **1. Missing related works**
>
> Thank you for highlighting this relevant work. We agree these restoration prior methods [1-3] are important to discuss and have added these works to the paper.
> The key distinction is that RAM achieves non-iterative reconstruction, requiring only a single forward pass (versus the iterative optimization in [1-3]). Additionally, while [1-3] use backbones trained on single tasks (e.g., SISR), RAM is trained jointly on multiple tasks, enabling zero-shot generalization to new inverse problems without iterative adaptation.
> For problems that differ significantly from training tasks, we use self-supervised fine-tuning rather than iterative algorithms. We'll clarify this positioning in the related work section.
>
> **2. Reference on multiscale conditioning**
>
> Thank you for this reference, we have added it to our revised version.
> While both approaches use multiscale conditioning on measurements, RAM differs in two important ways:
> 1. Architecture: We upsample features to match the resolution of $A^{\top}y$  before Krylov iterations, then downsample and concatenate with measurements for encoding. This differs from [4]'s structure-guided approach.
> 2. Theoretical motivation: Our design stems from operator stability analysis, showing that forward operators exhibit better conditioning on coarse grids (see Fig. 3 of the paper for an illustration on how the proximal operator becomes more stable at coarser grids). We use sinc filtering during downsampling to prevent aliasing, which is critical for this stability property.
>
> Note that Table 4 (left, 3rd row) shows results with conditioning only on $A^{\top}y$, which is close to the approach in [4] and performs worse than our full multiscale design.
>
> **3. Conflicting signals at train time**
>
> We agree that some tasks can show conflicting signals during training. For instance, inpainting must add signal to the input image (i.e., add energy), while denoising tends to be preserve energy.
> However, our core hypothesis is that appropriate operator conditioning enables the model to route between these different behaviors depending on the inverse problem. The slight performance difference with added KSM blocks in Table 4 (left) suggests this works reasonably well, though we acknowledge specialist models could maximize performance for specific tasks.
>
> Yet, we believe that small performance gaps on individual tasks in exchange are outweighted by a single model that handles diverse inverse problems. This generality is particularly valuable for zero-shot transfer to new inverse problems with a small training dataset (see e.g. Fig. 5), where the multi-task training provides richer inductive biases than any single-task model could. We agree that a more detailed study of task interactions would strengthen the work and will consider this for future analysis.

---

### Author Response · Authors · 2025-11-21
**General response to reviewers**

We sincerely thank all reviewers for their thorough and constructive feedback. Your suggestions have substantially strengthened the paper. Below we summarize our revisions, followed by detailed responses to each reviewer. All changes in the updated manuscript are highlighted in red.

**Major Revisions**
New experiments addressing reviewer concerns:
1. *Runtime analysis* (Table 9): RAM is 3.7× faster than the unrolled baseline (uDPIR) and 120× faster than the diffusion baseline (DDRM) on 512×512 images
2. *PromptIR comparison* (Table 6, Appendix): Demonstrates that our proposed physics-informed conditioning provides +8dB improvement over prompt-based approaches such as PromptIR.
3. *Phase retrieval* (Section 5.4): We show zero-shot generalization to phase-retrieval problems via iterative linearization.
4. *Blind deblurring on real data* (Section 5.4, Appendix G): Added results on real blind motion blur dataset (Kohler et al., 2012), by pairing our model with a pretrained space-varying kernel estimation network (Carbajal et al., 2023).
5. *Extended related work*: Added restoration prior methods [1-3] and multiscale conditioning references [4] (as per reviewer bJqQ).

**Clarifications and improvements:**

1. *Added Section 3.1*: Clearer algorithmic motivation and theoretical grounding for the Krylov Subspace Module (KSM).
2. *Terminology*: Replaced "foundation model" with "general-purpose model" throughout to avoid confusion with large-scale multimodal pretraining.
3. *More baseline comparisons*: Clarified DDRM implementation details and statistical significance discussion, and comparison with SRNO model for super resolution.

Overall, we hope that three main messages emerge from our rebuttal:

1. RAM achieves competitive or superior performance to state-of-the-art across diverse inverse problems while being substantially more efficient (6× fewer FLOPs, 120× faster than diffusion methods).
2. A **new framework for physics-informed conditioning is our core contribution**, and is not an incremental architecture tweaking. We demonstrate that explicit operator conditioning outperforms both prompt-based methods (PromptIR) and algorithmic unrolling (uDPIR).
3. We show that a single model handles arbitrary linear operators and generalizes zero-shot to unseen non-linear tasks (blind deblurring, phase retrieval), which specialized methods cannot match.

We address each reviewer's specific concerns in detail below. We are committed to further revisions if any points remain unclear.

---

### Author Response · Authors · 2025-12-03
**Summary for the Area Chair**

Dear Area Chair,

Given that post-rebuttal interaction is not possible this year, we use this top-level comment to clarify key misunderstandings per reviewer and to highlight how the rebuttal (and updated manuscript) addresses the raised concerns.

## Overall assessment

Across all four reviews, three reviewers (bJqQ, sdNp, iwVF) expressed generally positive views with scores around the acceptance threshold, while one reviewer (uPNi) recommended rejection largely based on concerns that we believe are addressed by the rebuttal. The updated manuscript now includes: (i) clearer algorithmic motivation and positioning (new Section 3.1 and extended related work), (ii) **new experiments on blind deblurring and phase retrieval**, and additional comparisons with PromptIR and SRNO (iii) a detailed runtime analysis. **These experiments reveal** that RAM is 3.7× faster than unrolled baselines and 120× faster than diffusion while matching or surpassing them on 15+ diverse inverse problems, including several real-data settings.

Crucially, the rebuttal clarifies that the main contribution is not a minor architectural tweak of DRUNet **or unrolled architectures**, but a general framework for explicit operator conditioning that (1) yields a single, non-iterative, general-purpose model for a broad spectrum of computational imaging tasks, and (2) can be **easily** adapted to other backbone architectures, new modalities and imperfect operators. In light of these clarifications and additional results, we respectfully believe that the paper meets the bar for acceptance at ICLR.

----------------------------

### Reviewer bJqQ

Reviewer bJqQ overall **evaluated the paper positively (score 6/10), highlighting the strength of the KSM design, the breadth of experiments, and strong performance**. The main concerns regarding missing references and a potential trade-off between multi-task and task-specific training are summarized below:

- *Positioning with respect to the exisiting literature on restoration priors [1–3] and multiscale conditioning [4]*

In the revised manuscript, we explicitly discuss restoration priors [1–3] and clarify that RAM differs in two core ways: i) RAM performs non-iterative reconstruction with a single forward pass rather than iterative fixed-point schemes; and ii) RAM is jointly trained on multiple tasks (deblurring, MRI, CT, etc.) instead of using single-task backbones (e.g., SISR) as in [1–3]. As we emphasize in the rebuttal, our goal to show that a single general-purpose model can achieve strong performance and zero-shot generalization without iterative optimization. We also clarify that using a basic multi-scale conditioning on $A^\top A y$ (Table 4, left, 3rd row) is closer to [4] and performs worse than our full version.

- *Trade-offs of multi-task vs. task-specific training.*

Our key point is that the proposed multi-scale Krylov conditionning mitigates task conflicts exist (e.g., inpainting vs. denoising)  enables zero-shot transfer to new inverse problems, see Table 4 (+1.8 dB on cross-tasks).

----------------------------

---

> ### Author Response · Authors · 2025-12-03
>
> ### Reviewer uPNi
>
> The reviewer gave an **initial low score (2/10) despite having evaluated the paper positively**, stating that the paper provides "comprehensive experimentation (...) across diverse inverse problems (...) and domains" being a "valuable for benchmarking unified restoration models". The model achieves "performance comparable to unrolled and diffusion-based models while being roughly 8x faster and 10x smaller (36M parameters vs. >200M)" to existing baselines, "making it appealing for real-time or embedded applications.". They also state that "Demonstrating zero-shot and few-shot adaptation without ground truth (...) is practical" and the proposed model "can generalize with minimal data.", and regarding the presentation of the paper, "Despite the dense material, the manuscript maintains good structure and uses comprehensive figures".
>
> The low score was based on concerns, which we believe stem from misunderstandings that are carefully addressed in the rebuttal and revisions, and summarized below:
>
> - *Concerns about limited theory and ICLR fit*
>
>  Our contribution is not an unrolled iterative algorithm, but rather performs a single step inference. Convergence theory is thus irrelevant in our case, as it doesn't trivially apply to end-to-end networks. Nonetheless, we show how the proposed modules can generalize the way measurements are incorporated in existing deep unrolling models, and how the multiscale approach reduces the ill-posedness of the inverse problem. Moreover, we believe **there might have been a misunderstanding regarding the scope of ICLR**, since our contributions are analogous to recent ICLR imaging papers (Cui et al. 2023, 2025).
>
> - *Comparison with PromptIR/AdaIR/MambaIR*
>
> We clarified that while these methods focus on natural image restoration (dehazing, denoising, deraining, etc.), our approach explicitly conditions on the forward operator $A^\top A$, which is essential for scientific imaging (CT, MRI, SPAD, etc.). To demonstrate that this is not a minor tweak, we added a PromptIR comparison under identical training conditions (Table 6), showing that **our model yields up to +8 dB PSNR improvement over the PromptIR baseline**.
>
> - *“foundation” terminology*
>
> **We have replaced “foundation model” with “general-purpose model” throughout** to avoid confusion with large multimodal pretraining.
>
> - *Concerns regarding DDRM baseline*
>
> Finally, we clarified that the **modifications to the DDRM baseline were applied to maximize its performance**, as a naive application of the model would not provide competitive results due to its reliance on SVD-decomposable forward operators.

---

> > ### Author Response · Authors · 2025-12-03
> >
> > ### Reviewer sdNp
> >
> > Reviewer sdNp gave a marginally-below-accept score (4/10) but was overall **positive about the signal-processing grounding of the Krylov design and the clarity of the paper**, stating that the "paper showed promising results" and the "clarity of the contents is well defined".
> >
> > Their main concerns regarding comparison with iterative diffusion/promptIR/neural operators baselines were addressed in the rebuttal, showing that the proposed method significantly outperforms PromptIR, has significantly fewer parameters while keeping the same performance than diffusion baselines, and achieves similar performance while being applicable to a much larger set of inverse problems than SR neural operator baselines.
> >
> > - *Missing recent baselines (IPT/PromptIR, neural operators, diffusion for medical imaging)*
> >
> > We have significantly strengthened the comparison set. We now include PromptIR under identical conditions and show that while PromptIR and DRUNet are comparable as backbones, our physics-informed conditioning yields **gains up to +8 dB PSNR** for motion blur, and reduces hallucinations (Table 6, Fig. 8). We also **added comparisons to SRNO for super-resolution** (Fig. 12).
> >
> > - *Performance vs iterative diffusion models*
> >
> > We clarified that **RAM is competitive or superior to diffusion-based methods on almost all tasks**. The rebuttal explicitly notes that the only consistent case where DDRM slightly wins is Urban100 motion deblurring, while RAM outperforms diffusion baselines DDRM/DPS on virtually all other benchmarks. More importantly, runtime and complexity strongly favor RAM: the added Table 9 shows that RAM is ≈120× faster than DDRM (≈48 ms vs. ≈5.7 s), with an **order of magnitude fewer parameters (36M vs. 552M), and handles arbitrary operators** instead of SVD-limited ones.
> >
> > - *Lack of computational complexity analysis*
> >
> > We also **directly address complexity concerns around Krylov iterations**. As summarized in the rebuttal and Table 9, RAM’s matrix-free applications of $A$ and $A^\top$ scale similarly to conventional convolutions and do not introduce prohibitive overhead. In practice, RAM is **3.7× faster than uDPIR and 120× faster than DDRM** in wall-clock time.
> >
> > - *Comparison with SRNO/DiffFNO in severely degraded SR settings.*
> >
> > SRNO can slightly outperform RAM, but **it is specialized to super-resolution only, whereas RAM covers CT, MRI, deblurring, inpainting, and more**. Additional score-based methods for medical imaging use similar principles and would require bespoke setups; we explain this in the rebuttal.

---

> > > ### Author Response · Authors · 2025-12-03
> > >
> > > ### Reviewer iwVF
> > >
> > > Reviewer iwVF also gave a marginally-below-accept score (4/10) but **acknowledged the strength of RAM’s cross-modality generalization and the KSM’s novel way of embedding forward-operator knowledge**, stating that the model obtained "strong zero-shot and fine-tuned performance across a variety of modalities, often matching or surpassing heavier iterative or unrolled baselines while being computationally efficient." and "demonstrates generalization from a single training across multiple imaging modalities and degradation types."
> > >
> > > Their main concerns regarding a limited set of tasks (we clarify that more than 15 inverse problems are included in the paper), use of simulated data (we clarified that the paper includes various experiments on real measurement data with unknown noise models), lack of results in blind and non-linear settings (we added results on blind deblurrring and non-linear problems in the rebuttal), and differences with deep unrolling, were addressed in the rebuttal.
> > >
> > > - *The set of tasks and data was too limited/too simulated*
> > >
> > > We believe there was a misunderstanding regarding this point. **We evaluated our model on more than 15 inverse problems, and we included non-linear (phase-retrieval) and blind deblurring experiments in the rebuttal**. Importantly, **several experiments are on real measurements, not synthetic data**: fastMRI k-space acquisitions, real LinoSPAD measurements, real cryo-EM data, and real motion-blurred images (Köhler).
> > >
> > >
> > > - *Reliance on accurately known operators limited practicality*
> > >
> > > The rebuttal clarifies that **our method already addresses imperfect operators via self-supervised fine-tuning and blind/approximate operator settings**. For example, in LinoSPAD, the noise distribution is unknown and the faulty lines are not precisely known, yet RAM can be adapted using self-supervised losses that do not require a full noise model. Moreover, we included results during the rebuttal showing that RAM can be coupled with a kernel identification network to **perform competitive deblurring of real blurry images with unknown blur operators**.
> > >
> > > - *Multiscale approach limited to certain problems*
> > >
> > >  For multiscale conditioning, we clarify that the operator at multiple scales only requires matrix-free access to the forward operator. Thus, we do not need a special structure such as an SVD or explicit matrix form, and **our formulation applies to any operator for which matrix-vector products are available**.
> > >
> > > - *Novelty was incremental relative to deep unrolling*
> > >
> > > We updated Sections 3.1 and 3.2 to provide a clearer explanation that KSM generalises gradient and proximal steps from unrolled methods. Our design is motivated by optimization but deliberately abandons strict algorithm mimicry, showing that **what matters is how one conditions on the forward operator, not whether one reproduces an iterative scheme**, which we believe is an important discovery for the design of future reconstruction algorithms. Compared to deep unrolling, RAM offers **(i) better out-of-distribution robustness** on unseen noise models (Fig. 8), **(ii) stronger performance** on challenging problems such as 8× accelerated MRI (Fig. 7), and **(iii) substantially better efficiency** (≈6× fewer FLOPs and 3.7× faster runtime than uDPIR; Table 9) with a single-step architecture. We also clarify that unrolled networks like uDPIR, once parameters are learned and iterations fixed, do not retain formal convergence guarantees either. We demonstrate that explicit, learned operator conditioning within an image-to-image backbone can replace algorithm-mimicking iterations, yielding a single general-purpose model for diverse operators.

---

### Meta-Review · Area_Chair_6kLm · 2025-12-19

**Summary:**

The paper proposes the Reconstruct Anything Model, a non-iterative, lightweight architecture for solving various inverse imaging problems (CT, MRI, deconvolution, etc.) using a novel Krylov subspace module  for operator conditioning. Reviewers generally reached a consensus on the paper's strong empirical validation across diverse modalities and its significant computational efficiency, noting it is much faster than diffusion baselines while maintaining competitive performance. However, significant concerns were initially raised regarding the overstatement of the "foundation model" terminology, the perceived incremental nature relative to deep unrolling, and missing comparisons to recent methods like PromptIR or neural operators. The authors provided a substantial rebuttal that effectively mitigated these concerns by adopting the term "general-purpose" , providing detailed runtime analyses showing the method is orders of magnitude faster than DDRM, and adding other requested comparisons which demonstrated that explicit operator conditioning yields significant gains over prompt-based approaches. Furthermore, additional results on real-world data and non-linear tasks (i.e., phase retrieval) successfully addressed skepticism regarding the model's applicability beyond simulated linear problems.

**Reviewer Concerns:**

Addressed within the Rebuttal:
- The authors addressed the concern regarding the "foundation model" label by renaming it to "general-purpose model" throughout the manuscript.
- The authors added comparisons to PromptIR (showing +8dB improvement via physics-conditioning) and SRNO for super-resolution.
- In response to concerns about the cost of Krylov iterations, the authors provided a detailed runtime analysis demonstrating the model is $3.7\times$ faster than unrolled baselines and $120\times$ faster than DDRM.
- Concerns about reliance on simulated data and limited tasks were addressed by highlighting performance on 15+ inverse problems and adding experiments on real-world blind deblurring (Kohler dataset) and phase retrieval.

Outstanding or partially addressed:
- One reviewer requested formal convergence analysis. The authors argued this is irrelevant for a single-step inference model. While the empirical results are strong, the theoretical "guarantee" gap relative to optimization-based unrolling remains an inherent property of this approach that some reviewers might still weigh negatively.
- While the authors argue that explicit operator conditioning is a novel paradigm shift from, some reviewers could still understand the architecture as a clever engineering refinement of DRUNet rather than a fundamental breakthrough.

**Reviewer Scores:**

- Reviewer bJqQ (Intial Score: 6): Likely remains a 6/7. They were already positive, and their requests for missing references were addressed.
- Reviewer uPNi (Initial Score: 2): Could increase to a 4 or 5. The extremely low score was driven by the "oundation claim (which was addressed in the rebuttal) and runtime/baseline concerns, which were thoroughly refuted with the $120\times$ speedup data and PromptIR comparisons.
- Reviewer sdNp (Initial Score: 4): Could increase to a 5 or 6. Their primary concern was the lack of runtime analysis and missing neural operator baselines. The rebuttal provided the specific runtime table and SRNO comparisons requested.
- Reviewer iwVF (Current: 4): Probably increase to a 5 or 6. They were concerned about the small set of problems and simulated data. The authors demonstrated more than ten tasks, and added real-world Kohler and LinoSPAD experiments.

---

### Decision · Program_Chairs · 2026-01-26

Accept (Poster)